# On the usage of artificial intelligence in leprosy care: A systematic literature review

Hilson Gomes Vilar de Andrade [1,2], Elisson da Silva Rocha[1], Kayo H. de Carvalho Monteiro[1], Cleber Matos de Morais[3], Danielle Christine Moura dos Santos[4], Dimas Cassimiro Nascimento[1,5], Raphael A. Dourado[1], Theo Lynn[6], Patricia Takako Endo[1]*

1 Programa de Pós-Graduação em Engenharia da Computação (PPGEC), Universidade de Pernambuco (UPE), Recife, Brazil, 2 Instituto Federal de Educação, Ciência e Tecnologia de Pernambuco (IFPE), Recife, Brazil, 3 Programa de Pós-Graduação em Comunicação, Universidade Federal da Paraíba (UFPB), João Pessoa, Brazil, 4 Programa Associado de Pós-Graduação em Enfermagem (PAPGenf), Universidade de Pernambuco (UPE), Recife, Brazil, 5 Programa de Pós-Graduação em Ciencia da Computação, Universidade Federal de Campina Grande (UFCG), Campina Grande, Brazil, 6 Business School, Dublin City University (DCU), Dublin, Ireland

☉ These authors contributed equally to this work.
* patricia.endo@upe.br

**Data availability statement:** All relevant data are within the manuscript and its Supporting information files.

## Abstract

Leprosy, or Hansen's disease, is a Neglected Tropical Disease (NTD) caused by *Mycobacterium leprae* that mainly affects the skin and peripheral nerves, causing neuropathy to varying degrees. It can result in physical disabilities and functional loss and is particularly prevalent amongst the most vulnerable populations in tropical and subtropical regions worldwide. The persistent stigma and social exclusion associated with leprosy complicate eradication efforts exacerbate the wider challenges faced by NTDs in sourcing the necessary resources and attention for control and elimination. The introduction of Multidrug Therapy (MDT) significantly lowers the global disease burden. Despite this breakthrough in the treatment of leprosy, over 200,000 new leprosy cases are reported annually across more than 120 countries, emphasizing the need for ongoing detection and management efforts. Artificial Intelligence (AI) has the potential to transform leprosy care by accelerating early detection, improving accurate diagnosis, and enabling predictive modeling to improve the quality for those affected. The potential of AI to provide information to assist healthcare professionals in interventions that reduce the risk of disability, and consequently stigma, particularly in endemic regions, presents a promising path to reducing the incidence of leprosy and improving integration social status of patients. This systematic literature review (SLR) examines the state of the art in research on the use of AI for leprosy care. From an initial 657 works from six scientific databases (ACM Digital Library, IEEE Xplore, PubMed, Scopus, Science Direct and Springer), only 30 relevant works were identified, after analysis of three independent reviewers. We have excluded works due duplication, couldn't be retrieved and quality assessment. Results show that current research is focused primarily on the identification of symptoms using image based classification using three main techniques, neural networks, convolutional neural networks, and support vector machines; a small number of studies

**Funding:** This work was supported by the Conselho Nacional de Desenvolvimento Científico e Tecnológico (CNPq), Secretaria de Ciência, Tecnologia e Inovação e do Complexo Econômico-Industrial da Saúde (SECTICS), Ministério da Saúde (MS) grant 444509/2023-2 (to PTE). The funder had no role in study design, data collection and analysis, decision to publish, or preparation of the manuscript.

**Competing interests:** The authors have declared that no competing interests exist.

focus on other thematic areas of leprosy care. A comprehensive systematic approach to research on the application of AI to leprosy care can make a meaningful contribution to a leprosy-free world and help deliver on the promise of the Sustainable Development Goals (SDG).

## Author summary

In this study, we aim to pave the way for the effective use of an emerging technology that is increasingly integrated into our daily lives—Artificial Intelligence (AI)—to address a longstanding issue that has afflicted humanity for centuries and disproportionately affects the most impoverished populations: leprosy. Through a comprehensive and systematic review of the state-of-the-art applications of AI in the diagnosis, treatment, surveillance strategies, and epidemiological control of leprosy, we provide a detailed discussion of the solutions proposed to date, describing the materials and methods employed, reflecting on the reported limitations, and highlighting pathways for future advancements. These findings aim to enable early detection of new cases, interrupt transmission, develop new drugs, and prevent disabilities caused by the disease. Consequently, we understand that this work serves as a compass to guide the development of new AI-based technological solutions toward the global elimination of leprosy by 2030, as proposed by the World Health Organization.

## Introduction

Leprosy, also known as Hansen's disease, remains a significant global public health challenge despite advancements in detection and treatment strategies [1]. Leprosy is a Neglected Tropical Disease (NTD) caused by *Mycobacterium leprae*, which predominantly affects the skin, and peripheral nerves, causing neuropathy to varying degrees, which can result in physical disabilities and functional loss [2]. As an NTD, leprosy is prevalent in tropical and subtropical regions, affecting vulnerable populations with limited access to healthcare services. Historically, leprosy has been associated with significant stigma, social exclusion, and underreporting [3,4]. These factors not only complicate efforts towards eradication but also reflects the broader challenges faced by all NTDs in attracting the necessary attention and resources for comprehensive control and elimination strategies.

Recent data indicates that leprosy continues to occur in more than 120 countries, with over 200,000 new cases reported annually [5]. The World Health Organization (WHO) has made significant strides in leprosy control, particularly through the implementation of Multidrug Therapy (MDT), which has dramatically reduced the disease burden globally. However, new cases continue to emerge, signaling ongoing transmission and the need for sustained efforts in disease detection and management. In 2022, 174,087 new cases were recorded worldwide, with a significant concentration in the Region of the Americas, where 21,398 new cases were reported. Remarkably, 92% of these cases occurred in Brazil, highlighting the uneven geographic distribution of the disease and the need for targeted interventions in high-burden areas [6].

Artificial intelligence (AI) has the potential contribute to the eradication of leprosy and enhance the quality of life of those affected by leprosy through innovations in detection, care

and treatment. By facilitating early detection and providing accurate diagnoses, AI can significantly reduce the time to initiate treatment, the risk of long-term disabilities and the social stigma associated with visible symptoms. Furthermore, predictive modeling of transmission patterns enables targeted interventions in high-risk communities, potentially reducing the incidence of leprosy and disrupting cycles of transmission. Collectively, these advancements may contribute to a more hopeful prognosis for those affected, ensuring a better quality of life through improved health, increased social integration, and reduced discrimination.

The identification of scientific works at the intersection of leprosy care and AI can direct the development of new and innovative tools for detection, control and effective management strategies [7]. Recently, some reviews have been published analyzing the state of the art on the use of AI models to assist health professionals in leprosy-related decision making. For example, Fernandes et al. [8] reviewed works that used classic machine learning algorithms to develop models for diagnosing skin diseases, including leprosy. They restricted their review to the use of AI to the diagnosis of leprosy and mainly in differentiating leprosy from other diseases with dermatological manifestations, based on signs and symptoms. Similarly, Zinsou et al. [9] published a survey on works that used machine learning and deep learning for the early detection of leprosy, among other diseases, in black skin. In both cases, these reviews focus on a small number of topics in leprosy care focused primarily on identifying signs and symptoms of leprosy and diagnosis, excluding potential AI model applications in surveillance, treatment, healing and monitoring, and epidemiology.

Therefore, differently from the current literature, which predominantly discuss the application of AI in diagnosing leprosy, our systematic literature review (SLR) advances the discourse by focusing on underexplored dimensions. Specifically, we expand the analysis to include the integration of AI technologies within clinical workflows for leprosy care, emphasizing real-world applications and the operational challenges involved. Our SLR addresses this gap and in doing so provides a roadmap for researchers to explore the use of AI models in a more systematic way and contribute to WHO's Global Leprosy Strategy of zero leprosy. We explore the current landscape of research on AI models in leprosy care, describing findings on its effectiveness, challenges and future directions across the key thematic areas on leprosy care.

## Thematic areas in leprosy research

The purpose of an SLR is to identify, select and critically appraise research at the intersection of leprosy care and AI. Studies related to leprosy focus on a wide range of topics from exposure to the causative agent, the *Mycobacterium leprae*, to the transmission, development of symptoms, diagnosis, treatment, healing and epidemiology of leprosy. In contrast, extant reviews primarily focus on signs and symptoms and diagnosis only. The thematic categorization that will be used in this SLR is based on the clinical protocols and therapeutic guidelines divided into the following categories [10], as shown in Fig 1.

**Surveillance Strategy**: The main source of infection by the bacillus are untreated individuals affected by leprosy and with a high bacillary load, who expel *M. leprae* through the upper airways. Transmission occurs through direct person-to-person contact, and is facilitated by the coexistence of untreated patients with susceptible individuals. The incubation period of the disease is not precisely known, but it is estimated to last an average of five years [11].

Thus, contact surveillance is one of the main epidemiological surveillance strategies for early diagnosis and contact tracing of patients with leprosy, highlighting the active search of household contacts (HHC) as an important measure of disease control [12].

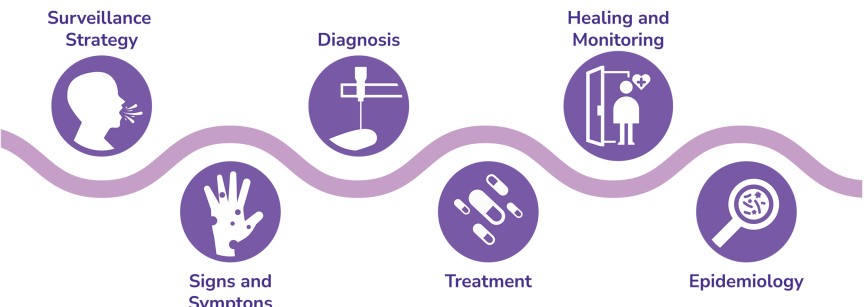

**Fig 1. Leprosy care thematic areas.**

**Signs and Symptoms**: Clinical symptoms and signs are used to define a leprosy diagnosis [13]. A case of leprosy is diagnosed by the presence of at least one or more of the following criteria: 1) lesion(s) and/or areas(s) of the skin with altered thermal and/or painful and/or tactile sensitivity; 2) peripheral nerve thickening, associated with sensory and/or motor and/or autonomic changes; 3) presence of *M. leprae*, confirmed by intradermal smear microscopy or skin biopsy [10].

**Diagnosis**: The best approach to avoid leprosy transmission relies on early diagnosis and treatment. Endemic countries often suffer from delays in diagnosis which result in disabilities for those affected [13]. The diagnosis of leprosy is mostly clinical involving a (complex) evaluation of skin lesions and peripheral nerves. In some cases, we can count on the support of complementary tests, such as: 1) direct smear microscopy for acid-fast bacilli (AFB); 2) histopathology; 3) ultrasound of peripheral nerves; 4) electroneuromyogram; 5) serological tests and 6) molecular biology test.

**Treatment**: Once diagnosed, leprosy can be treated with MDT, a combination of antibiotics. Treatment duration varies depending on the type of leprosy (paucibacillary or multibacillary). Without timely treatment, leprosy can lead to permanent nerve damage, disability, and disfigurement. Part of the treatment strategy includes rehabilitation and surgery to manage disabilities and improve the quality of life.

**Healing and Monitoring**: Once a leprosy patient is discharged, leprosy treatment requires a multidisciplinary approach that includes medical treatment to cure the infection, physical therapy and surgical interventions to manage disabilities, as well as psychological and social support to address the stigma and discrimination associated with the disease. This continues until the patient is cured of leprosy and associated health effects which can continue even when the patient is cured.

**Epidemiology**: Controlling the spread of leprosy involves identifying and treating infected individuals promptly to stop transmission. Contact tracing and prophylactic treatment of close contacts are also essential strategies.

The integration of AI across all these thematic areas can increase their effectiveness and the overall management and eradication of leprosy. In addition to diagnosis [8], AI has the potential to identify high-risk areas, monitor treatment adherence, support rehabilitation efforts, and make the management of the leprosy cycle more efficient and targeted. Certain specific aspects of leprosy clinical management, such as the occurrence of inflammatory reactions during treatment and antimicrobial resistance, underscore the importance of utilizing AI to support leprosy treatment. This approach enables the implementation of personalized therapeutic strategies [14].

AI-driven personalized leprosy treatment begins with the comprehensive collection and integration of diverse patient data, including medical history, genetic information, clinical evaluations, prior treatments, and socioeconomic context. The remarkable ability of AI to process and analyze this extensive dataset provides a holistic understanding of the patient's condition, facilitating more precise and individualized treatment decisions [15].

## Methodology

The objective of this SLR is to present the state of the art of research on the use of AI in all thematic areas of leprosy. To accomplish this goal, we follow the PRISMA (Preferred Reporting Items for Systematic reviews and Meta-Analyses) flow diagram [16], as presented in Fig 2.

The SLR seeks to address the following Research Questions (RQ):

- (RQ1) Which leprosy thematic areas are being the AI research focus?
- (RQ2) What datasets are being used in AI-based leprosy research?
- (RQ3) Which AI models are being used in leprosy research?
- (RQ4) How is the performance of AI models used in leprosy research?

### Search strategy

The objective of the search strategy is to define the databases to be used and what kind of search will be performed. We selected the following databases for this study: ACM Digital Library, IEEE Xplore, PubMed, Scopus, Science Direct (Elsevier) and Springer. We conducted

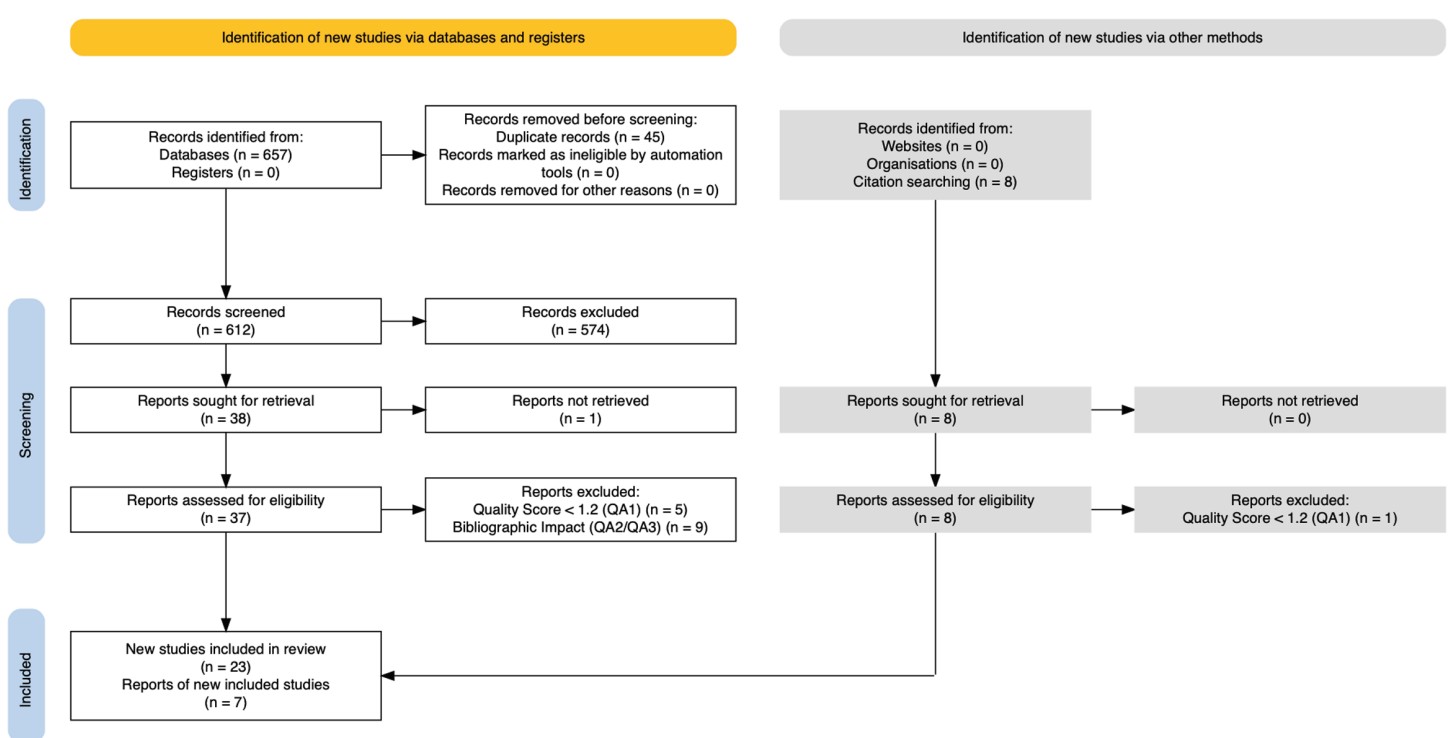

**Fig 2. PRISMA flow diagram.**

the manual search using Google Scholar (https://scholar.google.com) to identify and incorporate sources that might not be indexed in the selected databases.

To perform the automatic search in scientific datasets, we defined the search strings using a list of keywords based on the PICOC structure (Population, Intervention, Comparison, Outcomes, and Context) [17] and the SLR's objectives, with the purpose of guaranteeing that relevant terms would not be omitted. The final search string used was: ((leprosy **OR** hansen's disease) **AND** (artificial intelligence **OR** deep learning **OR** machine learning))

## Study selection

To ensure that only relevant studies were selected for this review, we established a set of inclusion (I) and exclusion (E) criteria. The inclusion criteria were: peer-reviewed articles selected from the repositories ACM Digital Library, IEEE Xplorer, Pubmed, Science Direct, Scopus and Springer Link on the use of artificial intelligence to address key aspects in leprosy care, as defined by the World Health Organization [7] (*I1*); and relevant studies cited by the works selected based on I1 (forward snowball search) [18] (*I2*). The exclusion criteria were: inaccessible works and those not featured in the selected databases (*E1*), works with only interpretations and evaluations of primary sources, classified as surveys (*E2*), duplicate works (*E3*), works not aligned with our RQ's (*E4*), and works not written in English (*E5*).

The initial automated search returned 657 records, which were consolidated into *.bib* files and exported to the Parsifal online tool [19], where forty-five duplicated records were removed and eight works selected by a single backward snowballing procedure, as they did not use automation tools, resulting in a set of 612 records. Next, two independent authors evaluated the work' titles and abstracts against the defined inclusion and exclusion criteria. Conflicts in inclusion/exclusion decisions were arbitrated by a third author. At the end of this process, 37 works remained. Finally, as will be detailed in the Quality Assessment section, we applied a sixth exclusion criteria (*E6*), which led to the exclusion of fourteen articles and inclusion of seven articles from citation searching, resulting in a final sample of 30 works. A numbered table of all studies in the literature search was included as S1 Table.

## Quality assessment

The objective of this activity is to define the criteria to measure the quality of each primary study. However, there is not an agreed definition of what a high-level quality study is; there is, though, a common agreement that the quality of the selected primary studies is fundamental to obtaining more reliable results [20].

Thus, we defined three quality assessment criteria (QA1-QA3) to be considered when applying the exclusion criteria *E6*, using an approach similar to that in Souza et al. [21] and based on bibliometric impact information. While QA1 uses four general and four specific criteria, QA2 uses the works' citations and QA3 relaxes QA2, as proposed by Duarte et al. [22].

QA1 is calculated using the *Quality Score* given by Eq 1, where the General *G* and Specific *S* assessment factors are summarized according to the following nested list:

- **General items (G) = 25%**
  - G1: Problem definition and motivation of the study:
    * (1) Explicit definition (1,0)
    * (2) General definition (0,5)
    * (3) No definition (0,0)
  - G2: Research methodology and organization:

* (1) An empirical methodology (1,0)
* (2) A generalized analysis (0,5)
* (3) Lacks any proper methods (0,0)
- G3: The study contributions refer to the study results:
    * (1) Explicitly correlates contributions to results (1,0)
    * (2) There is not correlation between contributions and results (0,5)
    * (3) No definition (0,0)
- G4: Limitation and future implications of the study:
    * (1) Formalized evaluation (1,0)
    * (2) Some informal evidences are provided (0,5)
    * (3) Non justified or *ad-hoc* validation (0,0)
- **Specific items (S) = 75%**
    - S1: There is an evaluation defined in the study:
        * (1) Formalized evaluation (1,0)
        * (2) Some informal evidences are provided (0,5)
        * (3) Non justified or *ad-hoc* validation (0,0)
    - S2: There is an experiment defined in the study:
        * (1) Formalized experimentation method (1,0)
        * (2) Some informal evidences are provided (0,5)
        * (3) Non justified or *ad-hoc* experimentation method (0,0)
    - S3: There are metrics to validate comprehension characteristics:
        * (1) Formalized definition of metrics (1,0)
        * (2) Some informal definition of metrics (0,5)
        * (3) Non justified or *ad-hoc* definition of metrics (0,0)
    - S4: There is use of another technique in addition to artificial intelligence:
        * (1) Formalized definition of another technique (1,0)
        * (2) Some informal definition of another technique (0,5)
        * (3) No definition (0,0)

The result is a numerical quantification to rank the selected studies. The quality assessment checklist, with $G$ and $S$ composed of four items each and each one with a maximum score of 1, produces a weighted average, where $S$ weight three times more than $G$, as the specific contribution *(S)* of a study is more important than the general contributions *(G)*. works with an overall score $\geq 2.5$ were considered "high" quality studies, works with a score $\geq 1.2$ and $< 2.5$ were considered "medium" quality and works with a score $< 1.2$ were considered of "lower" quality and were excluded from the analysis. It is important to highlight that there is no evaluation of the works' quality itself with this criterion, but only the contributions' alignment with this study's purpose.

$$QualityScore = (\sum_{n=1}^{4} G)/4 + [(\sum_{n=1}^{4} S)/4) * 3] \tag{1}$$

The second quality assessment criteria (QA2) rates works according to their citations. works with more than five citations received a "high" score, while the ones between 1 and 5 citations received a "medium" score, and works without citations a "low" score. We used Google Scholar to retrieve the number of citations for each work.

However, applying QA2 can be unfair to recent work, which will naturally have fewer citations. For these cases, the third quality assessment criteria (QA3) analyzes articles from the last five years that have potentially "high" relevance, have at least one citation and articles that have not been cited which have potentially "medium" relevance. For a work to be included in

the review, an article must obtain score $\geq 1.2$, and its criteria for bibliographic impact QA2 and QA3 must be "medium" or higher.

### Data extraction and coding

We extracted the following data for each study: authors, publication year, thematic area of leprosy care, dataset characteristics, artificial intelligence technique(s) used, and evaluation metrics.

## Results

Table 1 presents the overview of the data extracted from the works in the SLR sample to answer the RQs. In the following sections, we detail the answers to these questions.

### Leprosy thematic areas addressed

As shown in Fig 3, only the Healing and Monitoring area was not covered by our SLR sample. The majority of works (n=16) are related to the Signs and symptoms area, proposing the use of images of lesions on the skin to carry out multiclasses classification to identify different diseases, including leprosy [23,26,28,31,36,38,40,46–51,56–58].

Seven works are related to the area of Diagnosis, proposing: (a) the identification of leprosy through binary classification (leprosy/non-leprosy) [33,35,45]; (b) the identification of the disease's operational classification (paucibacillary (PB)/multibacillary(MB)) [27]; (c) the identification of genetic characteristics of leprosy patients [44]; (d) the diagnosis of subclinical leprosy cases, based on the patient's ability to induce the production of IFN-$\gamma$ [54]; and (e) to invesigating Bacterial Volatilome [55].

Three studies have underscored the critical area of the Surveillance strategy in leprosy management by leveraging laboratory data from HHC to devise strategies aimed at interrupting the transmission of the disease [25,30,34]. While Marçal et al. [25] and Gama et al. [30] utilized decision trees and random forest models to analyze cytokine release and integrated molecular-serological data, respectively, Tió-Coma et al. [34] adopted a transcriptomic approach, identifying a 4-gene signature capable of predicting leprosy up to five years before clinical onset. While these works share a common focus on early detection through sophisticated data analysis, they differ on the specific biological markers and predictive models employed.

Two works proposed the use of the Kohonen self-organizing mapping algorithm to identify clusters with high leprosy incidence to improve epidemiological strategies [24,37]. Both works analyzed HHC data within specific regions to improve surveillance and interventions. The first study [24] focused on active searches in Santarém, Brazil, using PGL-1 serology to pinpoint high-risk areas needing early intervention. da Silva et al. [37] broadened the range of data used by incorporating socio-economic data and refining the clustering process to tailor public health responses more effectively. Together, these studies demonstrate the potential of data mining in optimizing leprosy management by identifying critical areas for targeted action.

Only two studies proposed the use of AI to assist in the treatment of leprosy. Portelli et al. [41] proposed the development of a computational predictor for rifampicin (one of the main medications used in the polychemotherapy treatment of leprosy) resistance. The tool, named SUSPECT-RIF, extends the typical rifampicin resistance determining region (RRDR), using structural-based machine learning approaches to predict resistance mutations of the *M. tuberculosis* rpoB gene. This findings in this study and the associated tool could improve

**Table 1. Overview of the data extracted from the selected articles in the SLR.**

| Primary studies | Year | Thematic area | Data type | Dataset size and source | AI technique(s) used | Evaluation Metrics |
|---|---|---|---|---|---|---|
| Nyatte et al. [23] | 2023 | Signs and symptoms | Image | 234 images from Akonolinga and Ayoshospitals in Cameroon and the internet | Artificial Neural Networks (ANN) optimized by Genetic Algorithms (GA) | MSE and accuracy |
| Da Silva et al. [24] | 2018 | Epidemiology | Tabular Data | 772 leprosy cases notified by SINAN from 2003 to 2013 | Kohonen Self-Organizing Maps (SOM) | Cluster centroid analysis |
| Marcal et al. [25] | 2022 | Surveillance strategy | Tabular Data | Analysis of blood samples collected from 160 people in Brazil | Decision Tree (DT) | ROC curve |
| Rafay et al. [26] | 2023 | Signs and symptoms | 4,910 images | 31 curated from Atlas Dermatology and ISIC | Convolutional Neural Networks CNN - (EffcientNet, ResNet and VGG) | Accuracy, Precision, Recall and F1-score |
| De Souza et al. [27] | 2021 | Diagnosis | Tabular Data | 174,871 leprosy cases notified by SINAN from 2014 to 2019 | Random Forest (RF) | Sensitivity and Specificity |
| Steyve et al. [28] | 2022 | Signs and symptoms | Image | 1,054 skin lesion images colected in Cameroon [29] | SVM optimized by Black Hole Algorithm (BHO) | Accuracy, Specificity, F-score and Sensitivity |
| Gama et al. [30] | 2019 | Surveillance strategy | Tabular Data | Analysis of blood and slit skin smears samples collected from 433 people in Brazil | Random Forest (RF) | ROC curve, Sensibility and Specificity |
| Mondal et al. [31] | 2020 | Signs and symptoms | Image | Records generated by Generative Adversarial Network (GAN) [32] | Convolutional Neural Networks CNN - (DensenNet) | Accuracy |
| Baweja et al. [33] | 2016 | Diagnosis | Image | 120 images from Dermnetnz repository and the web | Convolutional Neural Networks (CNN) | Accuracy |
| Tió-Coma et al. [34] | 2021 | Surveillance strategy | Tabular Data | Analysis of genetic characteristics of blood samples collected from 5,352 people in Bangladesh | Random Forest (RF) | AUC, Accuracy, Specificity and Sensitivity |
| Baweja et al. [35] | 2023 | Diagnosis | Image | Images collected from Dermnetnz repository and the web | Convolutional Neural Networks (CNN) | Precision, Recall, F1 Score and Accuracy |
| Yotsu et al. [36] | 2023 | Signs and symptoms | Image | 1,709 images of 506 patients from Côte d'Ivoire and Ghana | Convolutional Neural Networks CNN-(ResNet-50 and VGG-16) | Accuracy |
| Dutra da Silva et al. [37] | 2018 | Epidemiology | Tabular Data | 40 leprosy cases notified by SINAN, in the city of Santarem, in 2014 | Kohonen Self-Organizing Maps (SOM) | Not described |
| Jin et al. [38] | 2020 | Signs and symptoms | Image | 350 face images from Disease-Specific face dataset [39] | Convolutional Neural Networks CNN-(ResNet-50 and VGG-16) | Accuracy, Precision, Sensitivity, Specificity and F1-Score |
| De Goma et al. [40] | 2020 | Signs and symptoms | Image | 686 skin disease images from people in Philippines | SVM and Artificial Neural Network (ANN) | Precision and Recall |
| Portelli et al. [41] | 2020 | Treatment | Tabular Data | 42 clinical *M. leprae* mutations curated from the literature [42,43] | Linear classifiers, Decision Tree (DT), K-NN, SVM and Ensemble classifiers | Sensitivity, Specificity, F1-Score, Accuracy and Precision |
| Zhang et al. [44] | 2016 | Diagnosis | Tabular Data | Real genotype data from 706 leprosy patients and 514 control people, from China | Bayesian Network, Neural Network (NN), Logistic Regression and Regression Splines | AUC and Brier score |
| Barbieri et al. [45] | 2022 | Diagnosis | Image and tabular data | 1226 images and sociodemografic data from 228 leprosy patients, from Brazil | Convolutional Neural Networks CNN-(ResNet-50, Inception-v4), Elastic-net Logistic Regression | ACC, AUC, Sensitivity and Specificity |
| Pal et al. [46] | 2013 | Signs and symptons | Image | 876 images collected from 141 dermatology patients from School of Tropical Medicine, in India | SVM | Accuracy |

(*Continued*)

**Table 1**. (Continued)

| Primary studies | Year | Thematic area | Data type | Dataset size and source | AI technique(s) used | Evaluation Metrics |
|---|---|---|---|---|---|---|
| Jaikishore et al. [47] | 2021 | Signs and symptoms | Image | 1,524 images collected from Dermnetnz repository and the web | Convolutional Neural Networks CNN-(MobileNet, VGG-16, Inception,Xception) | Accuracy, Precision, Recall and F1-Score |
| Das et al. [48] | 2013 | Signs and symptoms | Image | 876 images collected from 141 dermatology patients from School of Tropical Medicine, in India | SVM | Accuracy |
| Banerjee et al. [49] | 2023 | Signs and symptoms | Image | 876 images collected from 141 dermatology patients from School of Tropical Medicine, in India | SVM | Accuracy |
| Beesetty et al. [50] | 2023 | Signs and symptoms | Image | 396 images collected from 151 dermatology patients, in India | Siamese-based Few Shot Learning (FSL) | Accuracy, Sensitivity and Specificity |
| Surasinghe et al. [51] | 2023 | Signs and symptoms | Image | 867 images from Dermnet web page (https://dermnetnz.org) | Convolutional Neural Networks CNN - (EfficientNet) | Accuracy, Precision, Recall, and F1-Score |
| Khan et al. [52] | 2017 | Treatment | Tabular Data | 396 838 protein sequences from universal protein Resources, (uniProt) database [53] | SVM | Accuracy, Sensitivity, Specificity, MCC and F1-Score |
| Martins et al. [54] | 2012 | Diagnosis | Tabular Data | Analysis of genetic characteristics of blood samples collected from 127 volunteers people in Brazil | Artificial Neural Network (ANN) | Accuracy |
| Beccaria et al. [55] | 2021 | Diagnosis | Tabular Data | Analysis of biological characteristics of seven mycobacteria species | Random Forest (RF) | MS Similarity |
| Monisha et al. [56] | 2019 | Signs and symptoms | Image | The dataset consists of more than 25,000 pictures of various types of sickness from ISIC (https://www.isic-archive.com) | Gaussian Mixture Model (GMM) and Probabilistic Neural Network (PNN) | Not described |
| Pattnayak et al. [57] | 2024 | Signs and symptoms | Image | 1709 images collected from 506 patients, in India | Convolutional Neural Networks CNN - (RestNet-50 and VGG-16) | Accuracy and MCC |
| Yasir et al. [58] | 2014 | Signs and symptoms | Image and tabular data | 775 images and sociodemographic data collected from 128 dermatology patients, in Bangladesh | Artificial Neural Network (ANN) | Accuracy |

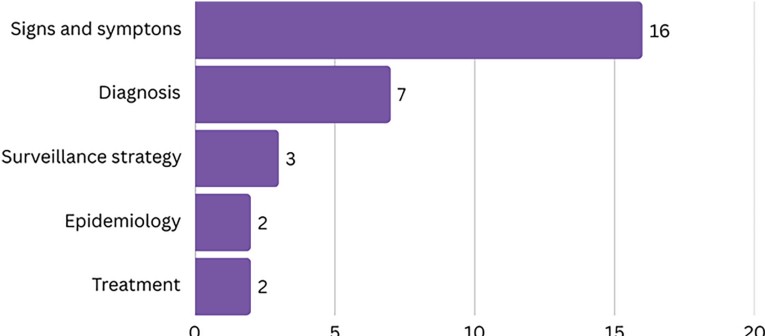

**Fig 3. Number of works in each thematic area of the leprosy care in the SLR sample.** The vertical axis lists the thematic categories identified: Signs and symptoms, Diagnosis, Surveillance strategy, Epidemiology, and Treatment. The horizontal axis represents the number of studies included in each category.

and accelerate the detection and management of drug-resistant leprosy, a critical step in the effective treatment phase of leprosy management. Khan et al. [52] analyzed an effective and precise approach for predicting mycobacterial membrane proteins. This work could contribute to the development of a potent tool for creating anti-mycobacterium drugs aimed at leprosy treatment.

The findings from RQ1 indicate that, although AI has been integrated into various aspects of leprosy management, there are considerable opportunities for further exploration. In all the thematic areas described, unexplored opportunities for the use of AI to advance leprosy care are observed. Whether in the combination of techniques or in the fusion of different types of data to develop new applications for early diagnosis, based on signs and symptoms, in addition to the use of images of skin lesions, or even in the development of new platforms to support clinical decision-making in the areas of treatment, healing and monitoring, based on supervised machine learning techniques, to predict, for example, the increase in the Grade of Physical Disability (GPD) induced by leprosy, or the occurrence of leprosy reactions.

Key areas for future research include refining treatment protocols and improving long-term patient monitoring within epidemiological studies. By capitalizing on these opportunities, we can address the current shortcomings in care and develop a more thorough and effective approach to leprosy management. This strategic enhancement promises to not only fill existing care gaps, but also elevates the overall quality of patient outcomes.

## Datasets used in AI-based research on leprosy

The availability, distribution and quality of datasets have become a crucial factor affecting the performance of machine learning models [59]. We analyzed three characteristics of the datasets used in the SLR sample: (a) type of data, (b) source of public datasets, and (c) data records and balancing.

**Type of data.** Data can be of various forms, such as structured, semi-structured, or unstructured [60,61]. Regarding the form, or type of data, the works in the SLR sample were divided into three categories: (a) image (where the data is presented in an unstructured way, without a pre-defined format), (b) tabular data (follows a standard order, being easier access and use by an entity or computer program) and (c) hybrid data, which consists of images and tabular data, as shown in Fig 4.

The majority of the thirty included works (n=16) utilized skin lesion images as input data for training models. Among these works, seven of them used images of skin lesions from public repositories [26,33,35,38,47,51,56], while the others used private datasets obtained from patients undergoing treatment in hospitals in leprosy-endemic countries in Africa [28,36] and Asia [40,46,48–50,57,58]. Nyatte et al. [23]combined the use of private and public skin lesion images. A single work proposed the use of facial images, from a public repository, for the early diagnosis of multiple diseases, including leprosy [38].

Eleven works developed research based on tabular data derived from laboratory analyses [25,30,34,41,44,52,54,55] and sociodemographic data [24,27,37]. Only two studies used hybrid data (image and tabular data). Barbieri et al. [45] proposed the combination of skin images classified as "leprosy-like lesions" (macule, plaques or nodules) with sociodemographic data (not detailed) to evaluate the binary classification (leprosy/non-leprosy) using neural networks. Also using artificial neural networks and images of lesions, Yasir et al. [58] proposed the inclusion of eight specific information (gender, age, duration, liquid type, liquid color, elevation and feeling) for the automatic detection of nine types of skin diseases, including leprosy.

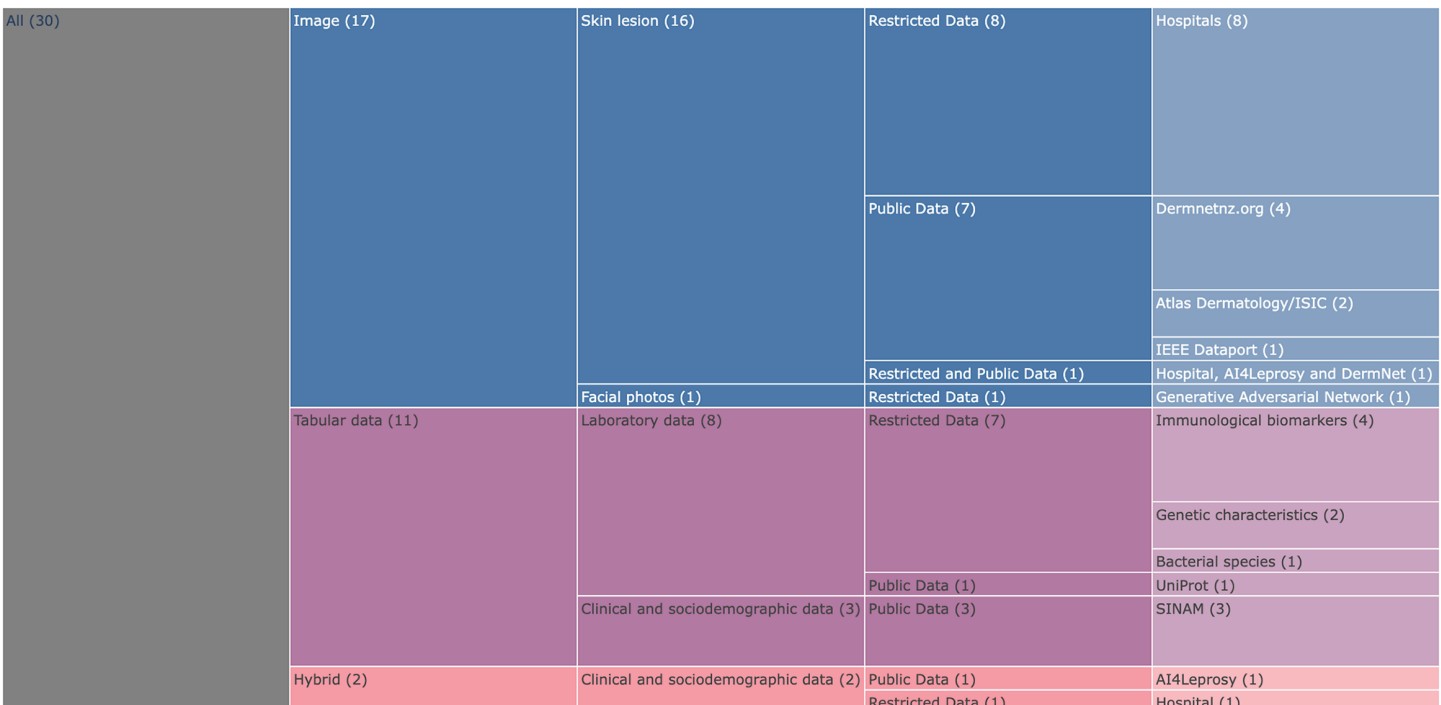

**Fig 4. Type and source of datasets used in the SLR sample.** The 30 studies were divided firstly in relation to the type of data: Images, Tabular and Hybrid. Within each type they were separated into data characteristics and data availability (public or private). Finally, the name of the data set used is presented in the last column.

**Source of public datasets.** We also analyzed the source of the data used in the SLR sample. The availability of data is considered the key to build machine learning models and data-driven real-world systems in general [62,63]. This information is crucial to allow the reproduction of the experiments described in each of them, as well as to make it possible to run new experiments applying other techniques to the same datasets – thus enabling the comparison of the results obtained on each scenario.

We identified two types of data sources: (a) restricted data, where no open source repository was indicated, and (b) public data, where the source repositories were indicated, allowing free access, use and replication. Table 2 summarizes all public datasets found in this SLR and their respective sources.

Dermnetnz and SINAN (from Portuguese, Sistema de Informação de Agravos de Notificação) are the most used public datasets. Dermnetnz is a resource for dermatological education and research, offering a wide range of clinical images, with more than 25,000 images; while SINAN is an essential resource for tracking health conditions in whole Brazilian territorial, known for its comprehensive coverage of epidemiological data (not only for leprosy, but for compulsory notification diseases).

ISIC (International Skin Imaging Collaboration) was used by two works, and similar to Dermnetnz, it also provides specialized open dataset primarily focused on dermatological imaging, with more than 503,955 public images. The Atlas Dermatology, which contains records of 561 skin diseases, was also utilized in one work in conjunction with the ISIC dataset.

**Table 2. Overview of the public dataset sources.**

| Works | Dataset | Records | Type | Source |
|---|---|---|---|---|
| Barbieri et al. [45] | AI4leprosy | 1,229 | Image (Skin lesion) | https://arcadados.fiocruz.br |
| Nyatte et al. [23] | | not described | | |
| Rafay et al. [26] | Atlas | 3,399 | Image (Skin lesion) | https://www.atlasdermatologico.com.br |
| Rafay et al. [26] | ISIC | 1,511 | Image (Skin lesion) | https://www.isic-archive.com |
| Baweja et al. [33] | Dermnetnz | 120 | Image (Skin lesion) | https://dermnetnz.org |
| Baweja et al. [35] | | not described | | |
| Jaikishore et al. [47] | | 1,524 | | |
| Surasinghe et al. [51] | | 867 | | |
| Nyatte et al. [23] | | not described | | |
| Jin et al. [38] | IEEE DataPort | 350 | Image (Facial) | https://ieee-dataport.org |
| Monisha et al. [56] | ISIC | Not described | Image (Skin lesion) | https://www.isic-archive.com |
| da Silva et al. [24] | SINAN | 772 | Tabular data | https://portalsinan.saude.gov.br |
| de Souza et al. [27] | | 174,871 | | |
| Dutra da Silva et al. [37] | | 40 | | |
| Khan et al. [52] | UniProt | 838 | Tabular data | http://www.uniprot.org/ |

AI4leprosy is a high-resolution image open dataset of skin lesions, focused on leprosy diagnosis, currently with 1,456 records available for download; and has been used by two studies.

Disease-Specific Faces (DSF) dataset is available in the IEEE DataPort repository, and it was utilized in a single work aimed at proposing a diagnostic approach for leprosy, among other diseases, as well as associated phenotypic and genotypic characteristics. For this purpose, the repository comprises images sourced from professional medical publications, websites, medical forums, and hospitals.

UniProt Knowledgebase (UniProtKB) is a dataset that provides high-quality protein sequences annotated with functional information, utilized in a single work. It contains approximately 246 million sequence records, derived from sources such as the International Nucleotide Sequence Database Collaboration (INSDC), Ensembl. and RefSeq.

**Data records and balancing.** With the exception of three studies that propose the use of Kohonen Self-Organizing Maps (SOM) for identifying regions with a high risk of leprosy transmission [24,37] or suggest the use of the Random Forest model for feature selection in the characterization of seven types of bacteria that cause tuberculosis or leprosy [55], all other selected studies (n = 27) use supervised machine learning models. In this technique, a fundamental aspect is the number of records used for the model training, as well as the strategies adopted to address the issue of sample imbalance among the classes used. Table 3 presents the classes considered in these studies, as well as the number of samples in each class and their proportion relative to the total number of records in the training dataset used by the models. De Souza et al. [27] and Mondal et al. [31] did not provide a detailed distribution of the records among the classes, mentioning only the total number of records considered (88,427 and 2,380, respectively).

An observed limitation is the lack of detail relating to strategies adopted to deal with imbalanced classes in the studies. Only three works used balanced classes (with the same number of samples per class) [33,35,40], and in six works, the division of data between the classes considered was not even mentioned [27,31,45,46,48,56], raising concerns regarding the generalizability and reproducibility of their findings.

**Table 3. Distribution of sample per classes**

| Work | Classes | Samples | Proportion |
|---|---|---|---|
| Yotsu et al. [36] | Buruli ulcer | 784 | 0.458 |
| | Leprosy | 131 | 0.076 |
| | Mycetoma | 32 | 0.018 |
| | Scabies | 389 | 0.227 |
| | Yaws | 373 | 0.218 |
| Jin et al. [38] | Beta-thalassemia | 70 | 0.201 |
| | Hyperthyroidism | 68 | 0.195 |
| | Down syndrome | 70 | 0.201 |
| | Leprosy | 70 | 0.201 |
| | Control | 70 | 0.201 |
| Nyatte et al. [23] | Buruli ulcer | 328 | 0.400 |
| | Leprosy | 287 | 0.350 |
| | Leishmaniasis | 205 | 0.250 |
| Marcal et al. [25] | Leprosy | 30 | 0.1875 |
| | Healthy controls | 69 | 0.43125 |
| | Household Contact | 61 | 0.38125 |
| Rafay et al. [26] | Basal Cell Carcinoma | 418 | 0.0851 |
| | Dariers | 96 | 0.0195 |
| | Epidermolysis Bullosa Pruriginosa | 96 | 0.0195 |
| | Hailey-Hailey Disease | 145 | 0.0295 |
| | Herpes Simplex | 88 | 0.0179 |
| | Impetigo | 110 | 0.0224 |
| | Larva Migrans | 140 | 0.0285 |
| | Leprosy Borderline | 145 | 0.0295 |
| | Leprosy Lepromatous | 293 | 0.0596 |
| | Leprosy Tuberculoid | 234 | 0.0476 |
| | Lichen Planes | 132 | 0.0268 |
| | Lupus Erythematosus Chronicus Discoides | 115 | 0.0234 |
| | Melanoma | 126 | 0.0256 |
| | Molluscum Contagiosum | 134 | 0.0272 |
| | Mycosis Fungoides | 116 | 0.0236 |
| | Neurofibromatosis | 86 | 0.0175 |
| | Papilomatoss Confuentes And Reticulate | 90 | 0.0183 |
| | Pediculosis Capitis | 82 | 0.0167 |
| | Pityriasis Rosea | 129 | 0.0262 |
| | Porokeratosis Actinic | 127 | 0.0258 |
| | Psoriasis | 132 | 0.0268 |
| | Tinea Corporis | 114 | 0.0232 |
| | Tinea Nigra | 118 | 0.0240 |
| | Tungiasis | 133 | 0.0270 |
| | Actinic Keratosis | 130 | 0.0264 |
| | Dermatofibroma | 111 | 0.0226 |
| | Nevus | 373 | 0.0759 |
| | Pigmented benign keratosis | 478 | 0.0973 |
| | Seborrheic Keratosis | 80 | 0.0162 |
| | Squamous Cell Carcinoma | 197 | 0.0401 |
| | Vascular Lesion | 142 | 0.0289 |
| De Souza et al. [27] | PB | Not detailed | |
| | MB | | |
| Steyve et al. [28] | Buruli ulcer | 420 | 0.399 |
| | Leprosy | 372 | 0.353 |
| | Leishmaniasis | 262 | 0.248 |
| Gama et al. [30] | Household Contact | 113 | 0.576 |
| | Leprosy case | 43 | 0.219 |
| | Endemic Control | 40 | 0.204 |
| Mondal et al. [31] | Leprosy | Not detailed | |
| | Tineaversicolor | | |
| | Vitiligo | | |
| | Normal Skin | | |

(*Continued*)

**Table 3.** (Continued)

| Work | Classes | Samples | Proportion |
|---|---|---|---|
| Baweja et al. [33] | Positive | 60 | 0.5 |
| | Negative | 60 | 0.5 |
| Tió-Coma et al. [34] | Progressors | 38 | 0.520 |
| | Control | 35 | 0.479 |
| Baweja et al. [35] | Positive | Not described | 0.5 |
| | Negative | | 0.5 |
| De Goma et al. [40] | Acne | 100 | 0.1666 |
| | Atopic Dermatitis | 100 | 0.1666 |
| | Keratosis Pilaris | 100 | 0.1666 |
| | Leprosy | 100 | 0.1666 |
| | Psoriasis | 100 | 0.1666 |
| | Warts | 100 | 0.1666 |
| Portelli et al. [41] | Resistant | 203 | 0.879 |
| | Susceptible | 28 | 0.121 |
| Zhang et al. [44] | Leprosy cases | 706 | 0.5787 |
| | Controls | 514 | 0.4213 |
| Barbieri et al. [45] | Leprosy cases | Not described | |
| | Controls | | |
| Pal et al. [46] | Leprosy | Not described | |
| | Tineaversicolor | | |
| | Vitiligo | | |
| | Normal Skin | | |
| Jaikishore et al. [47] | Measle | 41 | 0.336 |
| | Eczema | 988 | 0.8098 |
| | Leprosy | 126 | 0.1032 |
| | Normal Skin | 65 | 0.0532 |
| Das et al. [48] | Leprosy | Not described | |
| | Tineaversicolor | | |
| | Vitiligo | | |
| | Normal Skin | | |
| Banerjee et al. [64] | Leprosy | 262 | 0.2990 |
| | Tineaversicolor | 242 | 0.2762 |
| | Vitiligo | 210 | 0.2397 |
| | Normal Skin | 162 | 0.1849 |
| Beesetty et al. [50] | Leprosy | 368 | 0.9292 |
| | Non-leprosy | 28 | 0.0707 |
| Yasir et al. [58] | Eczema | 28 | 0.0361 |
| | Acne | 152 | 0.1961 |
| | Leprosy | 24 | 0.0309 |
| | Psoriasis | 99 | 0.1277 |
| | Scabies | 277 | 0.3574 |
| | Foot ulcer | 35 | 0.0451 |
| | Vitiligo | 62 | 0.08 |
| | Tinea Corporis | 66 | 0.0851 |
| | Pityriais Rosea | 32 | 0.0412 |
| Surasinghe et al. [51] | Cutaneous Leishmaniasis | 128 | 0.1849 |
| | Buruli ulcer | 132 | 0.1907 |
| | Leprosy | 127 | 0.1835 |
| | Mycetoma | 132 | 0.1907 |
| | Scabies | 173 | 0.25 |
| Khan et al. [52] | **dataset-I** | | |
| | Single pass | 32 | 0.1167 |
| | Multi pass | 192 | 0.7007 |
| | Lipids anchor | 20 | 0.0729 |
| | Peripheral membrane protein | 30 | 0.1094 |
| | **dataset-II** | | |
| | Membrane protein type | 274 | 0.4858 |
| | Non-membrane proteins | 290 | 0.5141 |

(*Continued*)

**Table 3**. (Continued)

| Work | Classes | Samples | Proportion |
|---|---|---|---|
| Monisha et al. [56] | Basal cell carcinoma | Not described | |
| | Scabies | | |
| | Zits | | |
| | Sickle-cell anemia | | |
| | Rubella | | |
| | Leprosy | | |
| | Psoriasis | | |
| | Measles | | |
| | Chickenpox | | |
| Martins et al. [54] | EC High | 20 | 0.1587 |
| | HCMB | 37 | 0.2936 |
| | HCPB | 27 | 0.2142 |
| | PB | 21 | 0.1667 |
| | MB | 21 | 0.1667 |
| Pattnayak et al. [57] | Leprosy | 131 | 0.0766 |
| | Yaws | 373 | 0.2182 |
| | Scabies | 389 | 0.2276 |
| | Buruli ulcer | 784 | 0.4587 |
| | Mycetoma | 32 | 0.0187 |

It is important to note that class imbalance affects metrics, such as accuracy, sensitivity (recall) and specificity. For instance, in highly imbalanced datasets, accuracy can be misleading, as a model may achieve high accuracy by simply predicting the majority class, while failing to correctly identify minority classes. In such cases, metrics like F1-score (which balances precision and recall) or AUC-ROC (which evaluates the model's ability to distinguish between classes across different thresholds) are more robust and informative.

Researchers should carefully select evaluation metrics that are appropriate for imbalanced datasets, and consider techniques such as synthetic minority over-sampling technique (SMOTE), adaptive synthetic sampling (ADASYN), and other forms of cost-sensitive learning to mitigate the effects of class imbalance.

To enhance scientific reproducibility and ensure robustness, it is crucial for future research to not only make data available and address class imbalance but also to clearly document all (data pre-processing) strategies employed in the study. Reporting these methods transparently will allow for the reproduction of results and validation of the models.

## AI models used in leprosy research

Fig 5 presents a categorization of AI models used by the literature, organized at different levels. Vertical categories represent additional subdivisions, while horizontal categories indicate the final categorization.

Initially, models are divided into two types of learning techniques: supervised and unsupervised learning. Within the supervised learning, there is a further subdivision regarding the target problem: classification and regression. Models used to solve classification problems are grouped into different types, including tree-based models, ensemble models, Support Vector Machines (SVM), neural networks, Siamese Networks, distance-based models, and linear classifiers. And models used for regression problems do not have additional subdivisions, making this their final categorization. Similary, unsupervised learing models fall exclusively under the clustering category, without further subdivisions.

More than 20 AI models were used, highlighting the diversity of approaches applied in the various research areas.

| Categorization | | | AI models |
|---|---|---|---|
| Learning Type | Supervised | Classification — Tree | Decision Tree (DT) |
| | | Ensemble | Random Forest (RF)<br>Extra Trees<br>XGBoost<br>Gradient Boost<br>AdaBoost |
| | | SVM | Support Vector Machine (SVM) |
| | | Neural networks | Artificial Neural Network (ANN)<br>Probabilistic Neural Network (PNN)<br>Neural Network (NN)<br>Convolutional Neural Network (CNN)<br>Generative Adversarial Network (GAN) |
| | | Siamese Network | Siamese-based Few-Shot Learning (FSL) |
| | | Distance | K-nearest neighbors (KNN) |
| | | Linear Classifiers | Multinomial and Complement Naïve Bayes<br>Bayesian Networks<br>Gaussian<br>Stochastic Gradient Descent |
| | | Regression | Logistic Regression<br>Regression Splines<br>Elastic-Net Logistic Regression |
| | Unsupervised | Clustering | Kohonen Self-Organizing Maps (SOM)<br>Gaussian Mixture Model (GMM) |

**Fig 5. AI models categorization.** This Figure presents the AI models categorized into different levels, starting with the type of learning: Supervised and unsupervised, where the vertical categories mean that there is still subdivision, and the horizontal categories represent the last category of models.

Fig 6 presents the distribution of AI models used in the literature per leprosy thematic area. It is important to highlight that the choice of AI models should be based on the nature of the data analyzed.

Regarding the thematic area, Signs and Symptoms was the area that had the largest number of trained models (18 works), with emphasis on the Convolutional Neural Networks (CNNs), Support Vector Machines (SVM) and Neural Networks (NN) models. Regarding the diversity of the models evaluated, Signs and Symptoms also stands out, but we would like to highlight the Treatment area, which with only two works [41,52] managed to test five different types of AI models.

Another important point to note is that the Kohonen Self-Organizing Maps (SOM) model was the only one used for Epidemiology [24,37]. This is because this is an unsupervised

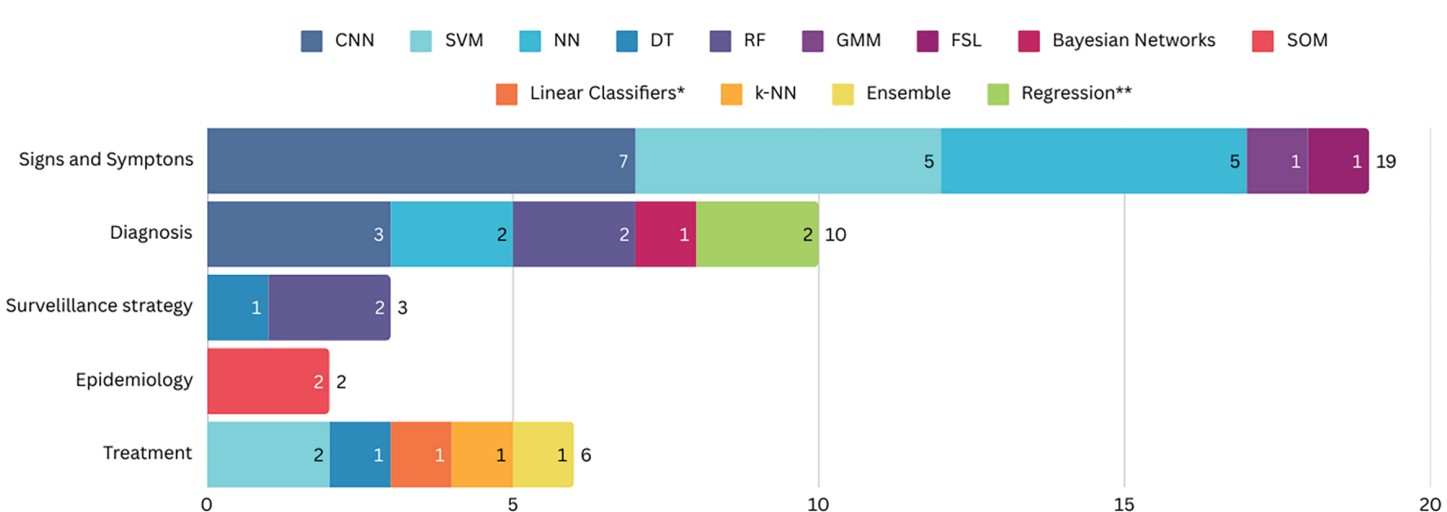

**Fig 6. Number of AI models per leprosy thematic area.** The vertical axis represents the thematic categories analyzed, while the horizontal axis represents the number of AI models included in each category. Each category is formed by several colors, where each color represents a specific model, detailed in the Figure legend. *Consider that Linear Classifiers correspond to Multinomial and Complement, Naïve Bayes, Bayesian Networks, Gaussian and Stochastic Gradient Descent models. **Consider that Regression correspond to the Logistic Regression, Regression Splines and Elastic-Net Logistic Regression models.

model, used for data clustering; this aligns with the epidemiological objective of grouping cases based on shared characteristics [65], thereby facilitating a better understanding of disease patterns across regions of interest.

However, not only the thematic area influences the choice of models. The type of data used is also of utmost importance in deciding which model to use. Fig 7 shows the models used according to the type of data.

CNNs have been most commonly used when the work involves image-based datasets, for visual analysis of skin lesions. CNNs are suitable for such tasks due to their powerful feature

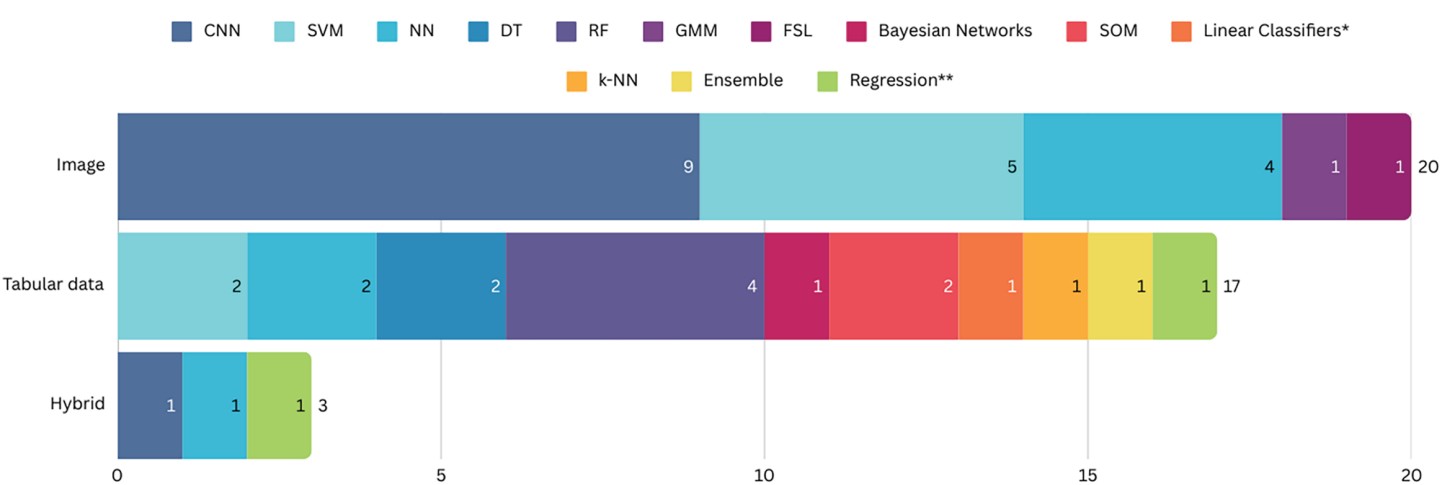

**Fig 7. Number of AI models per data type.** The vertical axis represents the types of data found (image, tabular and hybrid), while the horizontal axis represents the number of AI models included in each type. Each type is formed by several colors, where each color represents a specific model, detailed in the Figure legend. *Consider that Linear Classifiers correspond to Multinomial and Complement, Naïve Bayes, Bayesian Networks, Gaussian and Stochastic Gradient Descent models. **Consider that Regression correspond to the Logistic Regression, Regression Splines and Elastic-Net Logistic Regression models.

extraction capabilities in high-dimensional image data. SVM has also been a model widely used in the context of images. It is worth noting that in these studies, SVM dit not work alone, that is, other techniques are used as data preprocessing to assist in a given task, as is the case of the study by Pal et al. [46], who used SVM together with Rotation Invariant Weber Local Descriptor (WLD). The WLD was employed to improve feature extraction by capturing texture and edge information in a rotation-invariant manner, which is particularly useful for analyzing skin lesions with varying orientations and lighting conditions. This preprocessing step enhances the SVM's ability to classify images accurately. In contrast, tabular data models, including RF and DT, are more frequently used for surveillance and treatment prediction tasks, where interpretability is essential.

It is important to highlight that preprocessing techniques have a significant impact on model performance and generalization. In image-based approaches, for instance, variations in illumination normalization, lesion segmentation, or feature extraction methods—such as the inclusion or omission of descriptors like WLD—can lead to notable differences in model accuracy and generalizability. Similarly, in tabular datasets, preprocessing steps such as handling missing values, feature scaling, and encoding of categorical variables often vary considerably, directly influencing model inputs and outcomes. These variations underscore the importance of thoroughly documented preprocessing procedures to enhance reproducibility and enable fair comparisons across studies.

## Learning type

There is a clear predominance of supervised learning techniques in the classification and prediction of leprosy. Of the 30 works reviewed, 28 applied supervised learning methods, while only two were based on unsupervised learning approaches [24,37], as previously mentioned.

The main difference between supervised and unsupervised learning is the availability of labeled data. In supervised learning, the training dataset includes input-output pairs, where the output (label) is known. The model learns to map inputs to outputs, making it suitable for classification tasks such as lesion detection, disease diagnosis, and treatment prediction [66]. This explains the use of supervised learning in the reviewed studies, since leprosy-related tasks usually rely on well-defined labels, such as clinical diagnoses, lesion categories, or treatment outcomes. Common supervised models used include CNNs, SVMs, Random Forests, and NNs [67].

In contrast, unsupervised learning deals with unlabeled data. The model exploits the structure of the data to identify patterns, clusters, or relationships without predefined categories [66]. This approach is particularly useful for discovering hidden patterns in complex datasets, such as epidemiological analyses. In the works reviewed, [24,37] employed unsupervised learning techniques, including clustering methods, to analyze the transmission dynamics and epidemiological patterns of leprosy. These methods provided insights into how the disease spreads within communities, without relying on labeled datasets.

The prevalence of supervised learning in the reviewed literature indicates a focus on predictive accuracy and classification performance, which are essential for clinical applications.

## Convolutional neural networks

A Convolutional Neural Network (CNN) is a type of deep learning (DL) architecture specifically designed for the analysis of data structured as multi-dimensional arrays. A typical CNN consists of several key components: convolutional layers, pooling layers, non-linear activation functions (commonly the Rectified Linear Unit, or ReLU), and fully connected layers. In the

convolutional layers, neurons are arranged in feature maps, where each neuron is connected to localized regions of the feature maps from the preceding layer. This connectivity is achieved through a set of learned parameters known as filters.

The result of this whole process feeds fully connected layers resulting in a final classification [68]. Although the methodology of training and testing models is well-defined, the resultant models themselves can be often unexplainable to humans. Even when techniques are used to select attributes resulting in good model performance, the relationships between those attributes and the output classification may not directly track causal relationships in the real world [69].

CNNs were widely used for tasks involving image classification due to their superior performance in recognizing complex visual patterns. Models such as ResNet-50 and VGG-16 were applied by Yotsu et al. [36] and Pattnayak et al. [57] to classify skin lesion images, as leprosy, these models were selected for their ability to process clinical images collected from diverse regions and conditions. Baweja et al. [33] used Google's Inception-v3, capitalizing on its deep architecture that captures fine-grained features essential for leprosy lesion detection.

DenseNet was utilized by Mondal et al. [31] and Jaikishore et al. [47] in the context of signs and symptoms analysis due to its efficiency in information flow and resource reuse, addressing challenges related to small datasets and overfitting commonly found in lesion classification tasks. Beesetty et al. [50] developed a Siamese CNN model using Few Shot Learning (FSL) to deal with small datasets, improving model generalization and achieving 73.12% accuracy, which is relevant for early detection of leprosy where the amount of data is low. Jin et al. [38] applied deep transfer learning from face recognition tasks to facial disease classification, addressing small sample sizes through pre-trained networks to enhance diagnostic accuracy. Surasinghe et al. [51] combined EfficientNet-B3 with Grad-CAM visualization to detect skin diseases, including leprosy, the model achieved an overall classification accuracy of 91.53%, on the dataset created by the authors. Barbieri et al. [45] combined ResNet-50 and Inception-v4 models to classify confirmed cases based on embedded data. The hybrid model increased diagnostic accuracy and is adaptable for integration with smartphones.

## Neural networks

Inspired by the brain's ability to perform complex tasks, such as pattern recognition, while learning, memorizing, and executing motor control, algorithmic models based on biological neural systems have been proposed, referred to as artificial neural networks (NN). An artificial neuron (AN) is a model of a biological neuron (BN). Each AN receives signals from the environment, or other ANs, gathers these signals, and when fired, transmits a signal to all connected ANs. Input signals are inhibited or excited through negative and positive numerical weights associated with each connection to the AN. The firing of an AN and the strength of the exiting signal are controlled via a function, referred to as the activation function. The AN collects all incoming signals, and computes a net input signal as a function of the respective weights. The net input signal serves as input to the activation function which calculates the output signal of the AN. An artificial neural network (NN) is a layered network of ANs. An NN may consist of an input layer, hidden layers, and an output layer. ANs in one layer are connected, fully or partially, to the ANs in the next layer. Feedback connections to previous layers are also possible [70].

In general, NNs present many advantages including a high capacity to learn and generalize, and the ability to deal with imprecise, fuzzy, noisy, and probabilistic information [71,72]. As such, they are widely used in health research [73–75]. MLP was a popular ML solution in the

1980s with applications in various fields. Historically, MLP was considered a traditional ML model [76,77] however with the advent of DL, the conceptualisation of MLP was advanced and is now increasingly considered a form of DL [78].

Neural Networks were applied primarily to manage large image datasets. Yasir et al. [58] applied a feed-forward neural network using a hybrid dataset comprising clinical images and textual data, including features such as lesion elevation, fluid type, and duration, achieving 90% accuracy in classifying nine dermatological diseases, including leprosy.

Martins et al. [54] utilized NNs for genetic marker analysis, supporting genetic-based diagnostic approaches. De Goma et al. [40] compared the performance of a NN model against SVM for the classification of skin diseases, including leprosy, and the NN showed superior recall.

## Support vector machines

Based on Vapnik's statistical learning theory [79], SVM builds hyperplanes in a multidimensional space to separate instances of different classes. The objective is to identify the optimal separating hyperplane while simultaneously maximizing the margin between the support vectors [79,80], robustness, due to the ease of dealing with data with outliers, is one of its main advantages. Although DT models offer the advantage of interpretability, a significant limitation of SVM models is their lack of transparency, particularly when working with high-dimensional datasets. Additionally, SVM models tend to be memory-intensive, which can lead to slower processing of large and complex datasets [79].

SVMs were selected for their high performance on small to medium-sized datasets and their ability to find optimal decision boundaries. Jin et al. [38] combined CNN and SVM, leveraging the strength of SVM in improving classification margins. Pal et al. [46], Das et al. [48], and Banerjee et al. [64] have also applied SVMs to image classification tasks, addressing the need for robust and interpretable models.

Steyve et al. [28] compared the performance of SVM with K-NN and Decision Tree, while Baweja et al. [35] conducted an analysis between Random Forest, CNN, and SVM for skin disease classification. Additionally, De Goma et al. [40] compared the performance of a Neural Network model against SVM for skin disease classification, with SVM serving as a benchmark model for evaluating classification accuracy.

SVM was also applied in treatment-related works. Portelli et al. [41] used SVM for predicting rifampicin resistance in Mycobacterium leprae beyond the RRDR region through a structure-based machine learning approach, improving drug resistance classification. Similarly, Khan et al. [52] employed SVM in the Unb-DPC framework to identify mycobacterial membrane protein types. These studies demonstrate the adaptability of SVM in addressing challenges related to treatment prediction and molecular characterization.

## Random forest

RF is an ensemble technique based on bagging that combines several DTs. It is built randomly from a set of possible trees with $K$ characteristics in each node. "Random" in this context means that, in the set of trees, each tree has an equal chance of being sampled. Multiple classification trees are obtained from bootstrap samples in order to calculate the final majority classification.

RF algorithms were widely adopted for tabular epidemiological and genetic data due to their robustness and interpretability. De Souza et al. [27] applied RF to large epidemiological datasets from SINAN. Gama et al. [30] and Tió-Coma et al. [34]) used RF to manage serological and genetic data, benefiting from RF's ability to handle missing data and high

dimensionality. Beccaria et al. [55] and Khan et al. [52] further demonstrated RF's adaptability in classifying mycobacterial species and membrane proteins. Baweja et al. [35] was the only work that used images and compared the performance of RF with CNN and SVM.

## Other models and approaches

Elastic-net Regression, XGBoost, and ensemble classifiers were used by Barbieri et al. [45] to integrate multi-modal data, showcasing flexibility in handling complex datasets. Portelli et al. [41] employed linear classifiers, Decision Trees, and K-NN, highlighting their suitability for predicting drug resistance. Zhang et al. [44] combined logistic regression and Neural Networks to analyze genotype data, offering insights into genetic predispositions.

Da Silva et al. [24] and Dutra da Silva et al. [37] used unsupervised clustering techniques for pattern discovery in transmission dynamics. A DT based algorithm was applied by Marcal et al. [25] to classify leprosy patients and household contacts using cytokine biomarkers, addressing challenges in early diagnosis and disease classification. Additionally, GMM combined with a Probabilistic Neural Network (PNN) was used by Monisha et al. [56] for skin disease classification, employing preprocessing techniques like RGB to HSV conversion and texture feature extraction to classify skin diseases, including leprosy.

## Performance evaluation of AI models used in leprosy research

The metrics used to evaluate the performance of the models described in this research are based on the number of occurrences between the true classification and the classification predicted by the model [81]. It is composed of four values:

- TP: The number of values of the principal class that the model predicts right.
- FP: The number of values of the principal class that the model predicts wrong.
- TN: The number of values of the secondary class that the model predicts right.
- FN: The number of values of the secondary class that the model predicts wrong.

Fig 8 presents the metrics used to evaluate the models, by type of problems addressed in this research. We observed that eleven metrics were used to evaluate the performance of AI-based models in leprosy care. Among them, eight metrics were used in works whose models used for classification: 1) accuracy, 2) sensitivity (also referred to as recall), 3) specificity, 4) F1-score, 5) precision, 6) ROC curve and AUC, 7) MCC, 8) Brier score and 9) MSE; works where models were employed for clustering and feature selection used 10) cluster centroid analysis and 11) MS similarity as evaluation metrics. In two of the thirty selected studies, the metrics used to evaluate the proposed models were not described [37,56].

**Accuracy.** Accuracy is one of the most common evaluation metrics used to evaluate the generalization ability of the trained classifier i.e., to measure and summarize the quality of trained milticlass classifier [23,26,28,31,36,38,46–52,57,58] and blinary classifiers [27,33–35,41,45,54], when tested with the unseen data [82]. Unsurprisingly, it was the most common metric used in the selected works. Accuracy is calculated as the sum of TP and TN divided by the total of samples, as shown in Eq 2.

$$accuracy = \frac{TP + TN}{TP + TN + FP + FN} \tag{2}$$

Among all 22 works that used accuracy as a metric, there was a balance in the number of samples in the classes used by the classifiers in only one work [35]. The impact of class

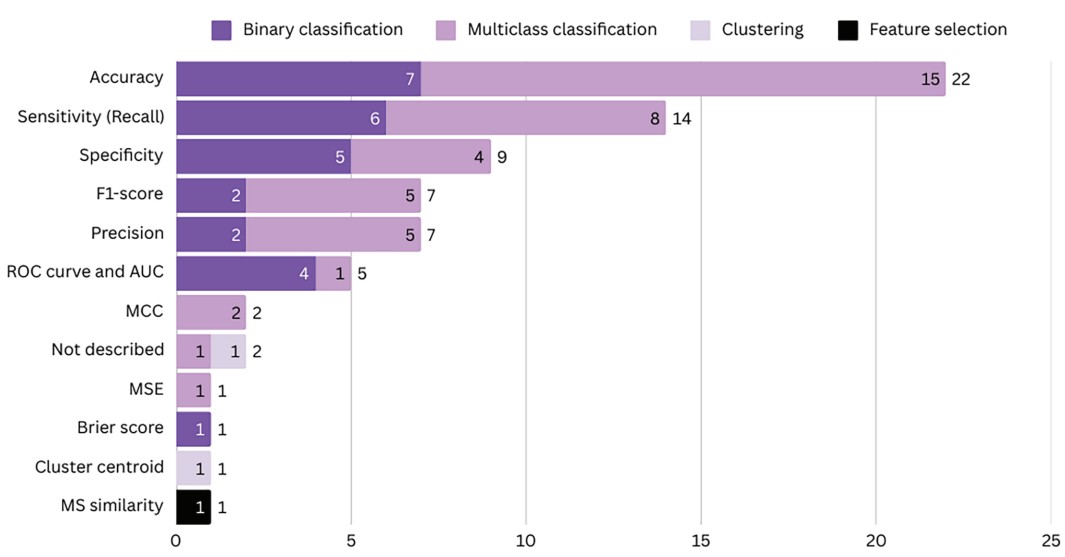

**Fig 8. Metrics used to evaluate the models by the type of problems addressed in the SLR sample.** The vertical axis presents the different metrics found in the study, while the horizontal axis represents the number of studies. The studies are further divided into colors representing the type of problem (Binary, Multiclass, Clustering and Feature selection).

imbalance on classification performance metrics is an important issue, especially in accuracy, according to systematic analysis presented by Luque et al. [83].

**Sensitivity or recall.**   Sensitivity (or recall) measures the fraction of positive patterns that were correctly classified by the model, critical for adoption of predictive models aimed at diagnostic applications in the medical field. For this reason, it is surprising that only 14 studies in the sample use it for evaluation, with eight works focused on multiclass classification [26,28,38,40,47,50–52] and six on binary classification [27,30,34,35,41,45]. It defines how well a model correctly predicted TP cases. It is calculated as the number of TP divided by the sum of TP and FN, as shown in Eq 3.

$$sensitivity = \frac{TP}{TP + FN} \tag{3}$$

**Specificity.**   As opposed to the sensitivity metric, the focus of evaluating specificity is to measure the fraction of negative patterns that were correctly classified by the model, again an important metric for health applications. Again surprising, it was used only by nine works, with five works focused on binary classification [27,34,40,41,45] and four on multiclass classification [28,38,50,52]. This metric determines how well the model correctly predicted TN cases. It is calculated by the number of TN divided by the sum of TN and FP, as per Eq 4.

$$specificity = \frac{TN}{TN + FP} \tag{4}$$

**Precision.**   Precision is used to evaluate the proportion of correctly predicted positive instances (TO) out of all instances predicted as positive, as per Eq 5. It is particularly relevant in scenarios where minimizing FP is critical, a consideration that has been uncommon in studies focusing on leprosy, where only seven works have employed this metric, with five

works focused on multiclass classification [26,38,40,47,51] and two on binary classification [35,41].

$$precision = \frac{TP}{TP + FP} \tag{5}$$

**F1-Score.** The F1-Score is the harmonic mean between two metrics: precision and sensitivity. It is used when the objective is to seek a balance between these two metrics, being calculated as presented in Eq 6. The F1-Score was used in seven selected works, five works focused on multiclass classification [26,38,47,51,52] and two on binary classification [35,41].

$$F1 - score = 2 \times \frac{precision \times sensitivity}{precision + sensitivity} \tag{6}$$

**ROC curve and AUC.** The ROC (Receiver Operating Characteristic) curve is a graph to analyse the discriminating ability of the model, that is, how well the model is able to divide between two classes. It is a graph with the TPR, the sensitivity, in the $x$ axis, and the FPR, the complement of the specificity, in the $y$ axis. Based on ROC, it is possible to calculate the AUC. Two works, one focused on multiclass classification [25] and the other on binary classification [30], used the ROC curve to evaluate the performance of the proposed models.

The AUC - Area Under the Curve, summarises the ROC curve in a single value, aggregating all the ROC thresholds. Its result varies between 0 and 1; an AUC of 0.5 represents a test without discriminating ability, while an AUC of 1.0 represents a test with perfect discrimination [84]. This metric were used in three selected works, all focused on binary classification [34,44,45].

## MCC

The Matthews Correlation Coefficient (MCC) is a metric particularly suited for evaluating the performance of classification models trained on imbalanced datasets [85], a characteristic prevalent in works focused on leprosy, being calculated as presented in Eq 7. However, surprisingly, this metric has been used in only two studies [52,57], focused on multiclass classification.

$$MCC = \frac{(TP * TN) - (FP * FN)}{\sqrt{(TP + FP) * (TP + FN) * (TN + FP) * (TN + FN)}} \tag{7}$$

**MSE.** The Mean Squared Error (MSE) estimates the performance of an algorithm in measuring the error in prediction between the effective return $Y_i$ computed ex-post and the value $\hat{Y}_i$ predicted by the algorithm [86], as shown in Eq 8.

$$MSE = \frac{1}{n} \sum_{i=1}^{n} (Y_i - \hat{Y}_i)^2 \tag{8}$$

Because it is widely used to evaluate the performance of regression models, including linear regression and polynomial regression, the MSE metric is less used in multiclass classification problems. Only selecter work used MSE: Nyatte et al. [23] proposed the adoption of a model based on NN (hybrid WOA-SSO-ANN model) to identify leprosy, Buruli ulcers or leishmaniasis from images.

## Brier score

The Brier Score is a strictly proper scoring function that corresponds to MSE, however, it is primarily designed for binary classification problems [87]. Zhang et al. [44] used this metric to compare the performance of three models aiming at the diagnosis of leprosy.

## Cluster centroid

As mentioned previously, clustering is the process of grouping data based on shared properties [65]. The performance evaluation of models employing this technique is conducted through the analysis of the distance between the data points in the dataset and the central point (centroid) of the cluster proposed by the model. The primary metrics used to measure this distance are the Euclidean distance, Manhattan distance or Minkowski distance, represented by Eqs 9, 10, and 11, respectively, where $p_i$ and $q_i$ represents the dataset points.

$$EuclidianDistance = \sqrt{\sum_{i=1}^{n}(p_i - q_i)^2} \tag{9}$$

$$ManhattanDistance = \sum_{i=1}^{n}|p_i - q_i| \tag{10}$$

$$MinkowskiDistance = \sum_{i=1}^{n}|p_i - q_i|^p \tag{11}$$

This metric was utilized by Da Silva et al. [24] to identify regions with delayed leprosy diagnosis, leveraging geographic data from sociodemographic variables and serological test results for the detection of antibodies such as phenolic glycolipid (PGL-1).

## MS similarity

The last metric mentioned by the selected studies was MS similarity. Although the authors did not provide details on how the presented values were calculated, it is observed that this metric was used to evaluate the performance of the RF algorithm for data feature reduction, aiming to identify different species of mycobacteria responsible for leprosy, among other infectious diseases [55]. By ranking features according to this score, RF offers a practical and intuitive approach to feature selection, enhancing the performance not only of RF itself but also of other machine learning models that may be affected by irrelevant or redundant features [88].

Table 4 provides an overview of the performance of AI-based models when employed to assist decision-making in leprosy care-related research. To ensure a fair comparison of the performance of these models, the metric (and its respective value) with the best-reported performance in each study was considered, along with the specific problem addressed by the work.

A trend toward the use of the accuracy metric for evaluating studies utilizing AI for multi-class classification can be observed, with results ranging from 73.12% [50] to 97.1% [52]. Certain limitations related to the datasets used in these works can be highlighted. While Beesetty et al. [50] employed a small dataset for training, comprising only 309 images collected from a single endemic region and without any initial preprocessing, Khan et al. [52],

**Table 4. Overview of reported results and limitations.**

| Work | Problem adressed | Performance | Reported limitations |
|---|---|---|---|
| Nyatte et al. [23] | Multiclass classification | Accuracy (87.45%) | Hardware restrict model complexity |
| Da Silva et al. [24] | Clustering | Cluster centroid | Requires expert interpretation |
| Marcal et al. [25] | Multiclass classification | AUC (0.8) | Needs validation across different regions |
| Rafay et al. [26] | Multiclass classification | Accuracy (87.5%) | Computationally expensive |
| De Souza et al. [27] | Binary classification | Accuracy (93.38%) | Inconsistencies in the dataset used |
| Steyve et al. [28] | Multiclass classification | Accuracy (96%) | High computational demands |
| Gama et al. [30] | Binary classification | Sensitivity (90.5%) | Specificity lower for PB cases |
| Mondal et al. [31] | Multiclass classification | Accuracy (87.5%) | Synthetic data may introduce biases |
| Baweja et al. [33] | Binary classification | Accuracy (91.6%) | Limited dataset may introduce biases |
| Tió-Coma et al. [34] | Binary classification | AUC (95.2%) | Limited validation sample size |
| Baweja et al. [35] | Binary classification | Accuracy (98%) | Needs validation across different clinical contexts |
| Yotsu et al. [36] | Multiclass classification | Accuracy (84.63%) | Limited dataset may introduce biases |
| Dutra da Silva et al. [37] | Clustering | Not described | Requires expert interpretation |
| Jin et al. [38] | Multiclass classification | Accuracy (93.3%) | Limited dataset may introduce biases |
| De Goma et al. [40] | Multiclass classification | Precision (96.55%) | Needs validation across different regions |
| Portelli et al. [41] | Binary classification | F1-Score (0.94) | High computational demands |
| Zhang et al. [44] | Binary classification | AUC (0.8392) | Requires large datasets for optimal performance |
| Barbieri et al. [45] | Binary classification | Accuracy (96.4%) | Requires diverse datasets for generalizability |
| Pal et al. [46] | Multiclass classification | Accuracy (87.36%) | Computationally expensive |
| Jaikishore et al. [47] | Multiclass classification | Accuracy (94%) | Data augmentation techniques may introduce biases |
| Das et al. [48] | Multiclass classification | Accuracy (89.66%) | Needs validation across different regions |
| Banerjee et al. [49] | Multiclass classification | Accuracy (91.38%) | Needs validation across different regions |
| Beesetty et al. [50] | Multiclass classification | Accuracy (73.12%) | Limited dataset may introduce biases |
| Surasinghe et al. [51] | Multiclass classification | Accuracy (91.53%) | Needs validation across different regions |
| Khan et al. [52] | Multiclass classification | Accuracy (97.1%) | Requires expert interpretation |
| Martins et al. [54] | Binary classification | Accuracy (96%) | Limited dataset may introduce biases |
| Beccaria et al. [55] | Feature Selection | MS similarity (94%) | Requires expert interpretation |
| Monisha et al. [56] | Multiclass classification | Not described | Does not present any model performance metrics |
| Pattnayak et al. [57] | Multiclass classification | Accuracy (84.17%) | Needs validation across different regions |
| Yasir et al. [58] | Multiclass classification | Accuracy (89%) | Limited dataset may introduce biases regions |

although using tabular data obtained from the Universal Protein Resources database [53], performed prior data preprocessing with the support of experts. This effort resulted in two benchmark datasets, namely dataset-I and dataset-II, which were used as input data for the proposed model, leading to a classifier with significantly superior performance.

Unsurprisingly, the nine studies focused on binary classification [27,30,33–35,41,44,45,54] reported higher accuracy values compared to those addressing multi-class classification. Noteworthy is the result achieved by Baweja et al. [35] (accuracy = 98%), which can be attributed to the use of a balanced dataset of pre-processed images. However, the study does not specify the total size of the dataset used or the methodology employed for cleaning the samples obtained through web scraping, which hinders its reproducibility.

Finally, we observed that works focused on clustering and feature selection provided only a superficial description of the evaluation metrics used, compromising both their reproducibility and their adoption as benchmarks.

## Discussions

Our examination on research using AI in leprosy care suggests that while AI is predominantly utilized in the diagnostic area (with 76% of works), its potential remains underexplored in other thematic areas in leprosy care such as Surveillance Strategy, Epidemiology, Treatment, and Healing and Monitoring. Even in the diagnostic area, including studies based on

signs and symptoms, there are few records of clinical trials that guarantee the use of tools in a clinical context. In all studies, there is no evidence that the proposed models can obtain the indicated results when used with data obtained in regions other than those that generated the data used in training. Proving the generalization capacity of AI-based models used in leprosy care is an important gap to be explored.

Our analysis of 17 studies reveals a strong emphasis on the use of AI for diagnosing leprosy through image-based techniques. Models such as neural networks, CNN, and SVM have been applied to differentiate leprosy from other skin diseases, underscoring AI's robust capability in diagnostic accuracy and potential for supporting health professionals in this task. However, there is a dearth of studies on the integration of AI beyond the diagnosis thematic area. A small number of studies (n=3) have ventured into employing AI for surveillance and monitoring strategies, where it could significantly impact the early detection of leprosy cases and the interruption of transmission chains. For instance, the use of machine learning models in analyzing complex datasets related to and from HHC demonstrates AI's potential to identify transmission patterns and endemic regions. Despite this, few studies consider this data source or use case.

Moreover, the application of AI in the Treatment, such as the development of models to predict drug resistance, presents a crucial advancement but remains a novel area for further exploration. The study presented by Portelli et al. is a prime example as how AI can contribute to predict rifampicin resistance, highlighting the need for more research focused on AI application in optimizing treatment protocols and strain identification. We found only three works that used AI to support Surveillance Strategies to interrupt leprosy transmission. In this thematic area, one of the key points is the early detection of leprosy in HHC associated with infected patients. To this end, the use of AI-based models has been used to identify biomarkers that signal the early contagion of leprosy based on blood samples and to identify the contagion of HHC based on the analysis of blood samples and slit skin smears (SSS). In addition to these works, only four works make it clear that it is possible to expand the use of AI for the Diagnosis of leprosy beyond images of skin lesions. This low number of studies using tabular data, such as sociodemographic information, clinical and genetic data, shows that there is a gap in the diversity of datasets as input for AI-based models applied to decision-making in leprosy care.

In addition to the source of data, the source code used to implement the proposed models is an important aspect for understanding and evaluating the work. However, only five studies made this information available, and in varying levels of completeness. Nyatte et al. [23] presented the pseudo-code used in pre-processing the models' input data (images), while Steyve et al. [28] and Jin et al. [38] described the pseudo-code used to optimize the classification algorithm and feature extraction, respectively. Mondal et al. [31] made available the code used for data augmentation, with the aim of reducing the imbalance of samples between the classes considered by the model and Rafay et al. [26] shared only the code used in the implementation of the prototype of the web application developed to validate the proposed model.

Another gap observed in relation to the use of AI-based solutions for leprosy care is the lack of research investigating the perception of healthcare professionals about the use of these tools in the clinical context. This perception is crucial to guide the development of new solutions for thematic areas related to leprosy care, or even to identify new thematic areas not covered, as well as to ensure that the proposed AI-based solutions are effectively implemented and used effectively for the benefit of leprosy patients. From the work analyzed in this SLR, we identified that there is a great unexplored potential in the use of AI-based models to support both the detection and treatment of leprosy in a more systematic way.

## Conclusions

This SLR presented an overview of current literature that applies AI-based models to support clinical decision-making across all thematic areas of the leprosy care. Considering the multiple specificities of the disease, this research focused on works that use such models to carry out classification/prediction. We identified that extant research focuses mainly on identifying signs and symptoms, based on images, with the aim of anticipating diagnosis and interrupting the transmission of the disease. While this supports one of the essential pillars for achieving WHO's goal of achieving zero leprosy by 2030, it is not a comprehensive approach and underplays the potential of AI in leprosy care. Furthermore, these studies apply a limited number of machine learning and deep learning techniques: NN, CNN, and SVM.

We observed a small number of studies using AI in post-diagnosis including treatment, post-cure monitoring, as well as in support of surveillance strategies and epidemiological studies. Based on these thematic areas, only six studies employ methods based on decision trees and ensemble classifiers, in addition to neural networks. These works used clinical and sociodemographic data in their proposals, collected from primary care health units, and genetic data, obtained from blood samples from patients and HHC from endemic countries, such as Brazil and Bangladesh. No articles were found that used any of the techniques mentioned to help monitor patients after healing, an important step in preventing post-treatment leprosy reactions and supporting the actions necessary to reduce the effects of disabilities resulting from the infection, which contribute to the stigma of this disease.

We suggest that having an efficient and comprehensive clinical decision support system on leprosy can improve the quality of the entire clinical process of this ancient disease, adding value to decision-making at all areas of the care cycle. It would also help in defining public health policies, given that it is a neglected tropical disease, improving the use of resources and the patient's quality of life as a whole. In this context, the use of AI-based computational models appears as a viable option to be the "brain" of these systems, considering the large volume of data available, as well as recent advances in machine learning techniques. However, this requires a sustained, focused, and systematic approach, which places treatment and follow-up after healing as a differentiator. This demands greater coordination and sharing of datasets, as well as greater details regarding model configuration, feature selection, and evaluation metrics.

Eradicating leprosy requires both the development of innovative drug treatments and the implementation of robust organisational strategies to mitigate its impact, including early detection, case management, and community outreach. Future research is required for both. While the transformational potential of AI and machine learning in drug discovery has been heralded by a wide range of stakeholders [89–93], the findings of this SLR indicate that their integration into leprosy drug discovery remains limited. Only two studies highlighted their potential. Portelli et al. [41] developed SUSPECT-RIF, a structure-based machine learning tool that predicts rifampicin resistance mutations in M. tuberculosis rpoB – a strategy that could be adapted to rapidly detect drug-resistant M. leprae. Khan et al. [52] proposed an effective computational approach for predicting mycobacterial membrane proteins, potentially paving the way for targeted anti-mycobacterial therapies. Recent advances in explainable AI (XAI) provide new opportunities to enhance drug discovery by improving the interpretability of machine learning models, enabling more reliable target identification, virtual screening, and drug repurposing [94]. Given the genomic similarities between M. tuberculosis and M. leprae, AI-assisted drug repurposing has identified promising candidates such as telacebec (Q203) and TB47 [95]. Additionally, AI-driven modelling of mycolactone biosynthesis, a key virulence factor in M. ulcerans, could offer new therapeutic targets

for both leprosy and Buruli ulcer [95]. These are not the only parts of the drug discovery value chain that can benefit from AI [92,93]. Accelerated lead optimisation, refined virtual screening methods, and enhanced clinical trial design all offer the potential to revolutionise therapeutic development in leprosy.

While our SLR suggests there is greater scholarly focus on the use of AI in leprosy research for the classification of signs and symptoms (n=16), diagnosis (n=7), surveillance (n=3), epidemiology (n=2) and treatment (n=2), the volumes remain low. As such, future research should expand the application of AI to deepen and broaden these aspects of leprosy treatment and care. One key area is the integration of diverse data sources—clinical, demographic, structured, and unstructured—to train and test more robust machine learning models. Additionally, the potential of deep learning and ensemble models remains largely untapped, particularly for addressing critical challenges in post-diagnosis management. Optimising AI models through feature selection and hyperparameter tuning can further refine predictive accuracy and clinical relevance. In addition, adopting a more comprehensive set of evaluation metrics will be essential to address imbalanced datasets, ensuring the reliability of AI-driven diagnostic tools. Finally, the development of accessible AI-powered decision support systems could transform leprosy care in remote regions, providing reliable diagnostic assistance even in settings with intermittent connectivity. By addressing these research gaps, AI has the potential to play a transformative role in leprosy management, from early detection to personalised treatment strategies.

## Supporting information

**S1 Table. A numbered table of all studies in the literature search.**
(PDF)

**S2 PRISMA 2020 checklist. The PRISMA 2020 statement comprises a 27-item checklist and an expanded checklist that details reporting recommendations for each item.**
(PDF)

**S3 Legends. A list of all captions used to identify the figures and tables.**
(TXT)

## Acknowledgments

We would like to thank Fundação de Amparo à Ciência e Tecnologia do Estado de Pernambuco (FACEPE), Instituto Federal de Educação, Ciência e Tecnologia de Pernambuco (IFPE) and Universidade de Pernambuco (UPE).

## Author contributions

**Conceptualization:** Hilson Gomes Vilar de Andrade, Theo Lynn, Patricia Takako Endo.

**Data curation:** Hilson Gomes Vilar de Andrade, Kayo H. de Carvalho Monteiro, Patricia Takako Endo.

**Formal analysis:** Hilson Gomes Vilar de Andrade, Elisson da Silva Rocha, Kayo H. de Carvalho Monteiro, Raphael A. Dourado, Patricia Takako Endo.

**Funding acquisition:** Patricia Takako Endo.

**Investigation:** Hilson Gomes Vilar de Andrade, Elisson da Silva Rocha, Patricia Takako Endo.

**Methodology:** Hilson Gomes Vilar de Andrade, Patricia Takako Endo.

**Project administration:** Patricia Takako Endo.

**Supervision:** Patricia Takako Endo.

**Validation:** Patricia Takako Endo.

**Visualization:** Raphael A. Dourado.

**Writing – original draft:** Hilson Gomes Vilar de Andrade, Elisson da Silva Rocha, Cleber Matos de Morais, Danielle Christine Moura dos Santos, Dimas Cassimiro Nascimento, Raphael A. Dourado, Theo Lynn, Patricia Takako Endo.

**Writing – review & editing:** Hilson Gomes Vilar de Andrade, Elisson da Silva Rocha, Kayo H. de Carvalho Monteiro, Cleber Matos de Morais, Danielle Christine Moura dos Santos, Dimas Cassimiro Nascimento, Raphael A. Dourado, Theo Lynn, Patricia Takako Endo.

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
