## [Decision Letter · Decision Letter 0]

9 Jan 2025

PCOMPBIOL-D-24-01734

On the usage of artificial intelligence in leprosy care: A systematic literature review

PLOS Computational Biology

Dear Dr. Endo,

Thank you for submitting your manuscript to PLOS Computational Biology. After careful consideration, we feel that it has merit but does not fully meet PLOS Computational Biology's publication criteria as it currently stands. Therefore, we invite you to submit a revised version of the manuscript that addresses the points raised during the review process.

Please submit your revised manuscript within 60 days Mar 10 2025 11:59PM. If you will need more time than this to complete your revisions, please reply to this message or contact the journal office at ploscompbiol@plos.org. Please include the following items when submitting your revised manuscript:

We look forward to receiving your revised manuscript.

Kind regards,

Yang Lu, Ph.D.

Academic Editor

PLOS Computational Biology

Benjamin Althouse

Section Editor

PLOS Computational Biology

**Journal Requirements:**

Potential Copyright Issues:

i) Figure 2. Please confirm whether you drew the images / clip-art within the figure panels by hand. If you did not draw the images, please provide (a) a link to the source of the images or icons and their license / terms of use; or (b) written permission from the copyright holder to publish the images or icons under our CC BY 4.0 license. Alternatively, you may replace the images with open source alternatives. See these open source resources you may use to replace images / clip-art:

ii) Table 1 appears to have been previously published in this paper “Batista Duarte R, Silva da Silveira D, de Albuquerque Brito V, Lopes CS. A systematic literature review on the usage of eye-tracking in understanding process models. Business Process Management Journal. 2020;27(1):346–367”. Please provide written permission from the copyright holder to publish this under our CC BY 4.0 license, or remove the figure / replace the image. Please note we do not recommend using standard request forms available on Publishers' websites, as they grant single use rather than republication under an open access license.

2) State what role the funders took in the study. If the funders had no role in your study, please state: "The funders had no role in study design, data collection and analysis, decision to publish, or preparation of the manuscript.".

7) As required by our policy on Data Availability, please ensure your manuscript or supplementary information includes the following:

**Reviewers' comments:**

Reviewer's Responses to Questions

Reviewer #1: Hilson Gomes Vilar de Andrade et al collected and studied the publications using AI to study leprosy. The authors categorize all studies into 6 themes regarding different aspects of leprosy research. The authors followed PRISMA and screened for 395 publications and finally got 25 papers. The authors extracted information from the 25 papers and showed their thematic area, source of data, ML model types, problem types and metrics used to evaluate models.

The manuscript could be better if the authors could address the comments below.

Major comments:

1. (line 123) The search string is “((leprosy OR ”hansen’s disease”) AND (”artificial intelligence” OR ”deep learning” OR ”machine learning”) AND (prediction OR classification)).” The results need to contain either “prediction” or “classification”, but it looks like these two words could not cover all aspects of ML studies on leprosy and will have bias on more diagnostic studies. Other keyword like “regression”, ”treatment”, “prognosis” or “monitor” are also supposed to be included.

2. (Line 139) When the two independent authors screened 376 papers, only 35 remains. What are the numbers of papers excluded by each of the 5 criteria (E1 to E5)? In E2, it excludes papers “less than six pages”, does this “six pages” include supplementary information or references or additional information or it only refers to main text? Why is it “six” pages?

3. (Figure 6) Supervised learning models are usually used to solve two major types of problems, classification and regression. It looks like the studies collected in this manuscript are only about “classification” but none of them are about “regression”, is it because the search string did not include “regression”, or is it because the authors view regression problems also as classification problems, or is it due to there is no research using any regression models? It is expected that regression models could play essential roles in leprosy studies in like disease severity evaluation.

Minor comments:

1. It is recommended to explain the meaning of “SLR” (although the reviewer could guess it’s “systematic literature review”) when it appears in the text for the first time. This is important since this abbreviation appears 42 times in the manuscript.

Reviewer #2: In this work, authors presented a systematic literature review about artificial intelligence in leprosy care. This review is well organized. The authors showed different types of data and the leprosy areas that AI researches mainly focused on. Authors also introduced fundamental concepts about individual AI techniques and metrics used in current AI models in leprosy. Authors claimed that this review can help researchers to explore the use of AI models in a more systematic way for leprosy care. However, there are some issues needed to be addressed to improve the quality of this review and make it distinguished from existing study.

Major:

1, In line 39, authors mentioned the limitation of another review of AI models in leprosy (Fernandes et al. PMID: 38202187), which is limited to the use of AI to diagnose leprosy. However, in this review, at least 18 out 25 AI models had been discussed in the another review. Fernandes et al. also briefly mentioned the datasets and metrics used for these AI models. Therefore, this manuscript needs to be improved to make it distinguished from existing study.

Although authors introduce the fundamental concepts of AI techniques and metrics more comprehensively, authors should focus more on individual AI models. For instance, in line 520 “Seven articles in the SLR sample considered the use of CNN - [24,29,31,34,36,43,46,49].”, authors simply summarized them into one sentence. But in fact, these methods use different CNN architecture which are worthy to mention and discuss. (descriptions of other models also have this issue such as SVM, NN and etc.)

2, This review can be improved to better present audience a detailed picture of AI models in leprosy care in terms of their capability, pros and cons.Beyond simply listing the individual methods and techniques, authors should also mentioned the underlying rationale of individual methods to resolve the challenges in leprosy care. Authors should also report the achievement or results (like accuracy, F1-score, recall, precision, …) of individual methods.

3, There are some missing of genetic population groups discussion and accessibility to public data in RQ2. According to PMID: 34136455, Leprosy is directly related to the social factors in the population groups such as race, ethnicity, or skin color. In line 273, authors mentioned some datasets are public. The public data is always valuable for the entire community. To improve the accessibility and impacts of work, authors should provide a link to these public source data.

4, In line 162, authors described the cut-off used to separate literatures into different quality levels (QA1). Why these cut-offs (>3, 1.2~3, <1.2) are reasonable?

5, The number of studies in Fig 3 is 25. But the total number in Fig 5 and Fig 6 is different. Authors should clarify this point.

Minor:

1, The authors should spell out the SLR when it firstly mentioned.

2, Authors should also mention the AI in drug discovery for leprosy as a future direction.

3, Authors mentioned the imbalanced dataset is an issue for accuracy. However, this issue also exists in other metrics (PMID: 25574450), author should better describe it to guide the community to use proper metrics.

4, All figures should have figure captions.

Reviewer #3: N/A

Reviewer #4: This manuscript presents a systematic literature review (SLR) analyzing artificial intelligence (AI) applications in leprosy care, covering diagnosis, surveillance, treatment, and epidemiology. The authors identify that most AI applications in leprosy focus on image-based diagnosis, revealing significant gaps in other care phases.

Strengths:

The manuscript follows PRISMA guidelines. The manuscript's research questions (RQ1–RQ4) align well with clinical and technological needs. The paper presents detailed comparative tables and figures across studies.

Major concerns:

- The manuscript follows an empirical research format rather than a computational review format. For example, as a review paper, there are typically no methodological details about the review process itself. The evaluation metrics section should be trimmed. Sections like "Data extraction and coding" detract from the core computational analysis, while the algorithmic analysis should be expanded as the main focus.

- The authors provide only basic descriptions of ML techniques without analyzing their computational characteristics (in contrast, there are detailed formulas for evaluation metrics, which are less important). There is no analysis of algorithm details or future applications of specific methods.

- The scope of the manuscript is unclear. While the authors attempt a systematic review of leprosy care, they fail to establish the necessity and uniqueness of the problem. There is no analysis of how this problem differs from other tasks and why solving and understanding it is important (for example, why leprosy image analysis is computationally unique). Although the authors make general points, these aren't specific to the problem setting. For example, from lines 97 to 101, the authors mention that AI could improve leprosy management but don't provide specifics.

- The authors simply describe previous research without critical analysis and miss discussing why certain methods succeed or fail. Additionally, there is no synthesis of emerging methodological trends. These factors make it more like a catalog of papers than a critical review. For example, at line 201, the authors describe a gene signature identified by a paper but don't explain what it is, its implications, or its relevance to the topic.

Minor issues:

- Line 160. Is there a typo? It says "where S weight three times more than S"

- Line 246, what's the difference between the three data types, specifically in this setting?

- Line 407, most ML methods model the probability of an event, then what is the conceptual difference?

- Line 391, 340, for such interpretability, how to read the results?

Recommendations:

Due to these concerns, I cannot recommend this manuscript for publication in PCB. I suggest the authors: 1. Review other PCB publications to correct the format. 2. Discuss the problem setting more thoroughly. 3. Extend the scope, reduce descriptive content, and expand high-level discussion.

**Have the authors made all data and (if applicable) computational code underlying the findings in their manuscript fully available?**

Reviewer #1: None

Reviewer #2: None

Reviewer #3: None

Reviewer #4: Yes

PLOS authors have the option to publish the peer review history of their article (what does this mean?). If published, this will include your full peer review and any attached files.

Reviewer #1: No

Reviewer #2: No

Reviewer #3: No

Reviewer #4: No

**Figure resubmission:**
---

## [Decision Letter · Decision Letter 1]

28 Apr 2025

PCOMPBIOL-D-24-01734R1

On the usage of artificial intelligence in leprosy care: A systematic literature review

PLOS Computational Biology

Dear Dr. Endo,

Thank you for submitting your manuscript to PLOS Computational Biology. After careful consideration, we feel that it has merit but does not fully meet PLOS Computational Biology's publication criteria as it currently stands. Therefore, we invite you to submit a revised version of the manuscript that addresses the points raised during the review process.

Please submit your revised manuscript within 30 days Jun 28 2025 11:59PM. If you will need more time than this to complete your revisions, please reply to this message or contact the journal office at ploscompbiol@plos.org. Please include the following items when submitting your revised manuscript:

We look forward to receiving your revised manuscript.

Kind regards,

Yang Lu, Ph.D.

Academic Editor

PLOS Computational Biology

Benjamin Althouse

Section Editor

PLOS Computational Biology

**Journal Requirements:**

2) Please upload figure 1 as a separate Figure file in .tif or .eps format. For more information about how to convert and format your figure files please see our guidelines:  

              https://journals.plos.org/ploscompbiol/s/figure

3) Please upload the figures in a correct numerical order in the online submission form. We noted that the flowchart is Figure 2 in the manuscript. Please note that the flowchart should be uploaded as Figure 1. Please ensure that the main figures are uploaded with the file type "Figure" not "Supplemental.”

4) We have noticed that you have uploaded Supporting Information files, but you have not included a list of legends. Please add a full list of legends for your Supporting Information files in the manuscript after the references list. 

5) We note that your Data Availability Statement is currently as follows: "All relevant data are within the manuscript and its Supporting Information files.". Please confirm at this time whether or not your submission contains all raw data required to replicate the results of your study. Authors must share the “minimal data set” for their submission. PLOS defines the minimal data set to consist of the data required to replicate all study findings reported in the article, as well as related metadata and methods (https://journals.plos.org/plosone/s/data-availability#loc-minimal-data-set-definition).

State the initials, alongside each funding source, of each author to receive each grant. For example: "This work was supported by the National Institutes of Health (####### to AM; ###### to CJ) and the National Science Foundation (###### to AM)."State what role the funders took in the study. If the funders had no role in your study, please state: "The funders had no role in study design, data collection and analysis, decision to publish, or preparation of the manuscript."

**Reviewers' comments:**

Reviewer's Responses to Questions

Reviewer #1: The authors have addressed all my concerns in details. There are no more questions before I recommend to accept this manuscript.

Reviewer #2: Thank authors for addressing my concerns. This version is more comprehensive and would serve as a good start point for researchers interested in studying leprosy care + AI.

Reviewer #3: Authors have addressed the previous reviewers questions and made adjustments, recommending for acception.

Reviewer #4: This revised manuscript presents a comprehensive systematic review examining the application of artificial intelligence across the spectrum of leprosy care. I think that the authors have thoroughly addressed a lot of concerns raised in my previous review. The manuscript now appropriately focuses on computational aspects rather than review methodology, with substantially expanded sections on AI model categorization, technical implementation details, and critical analysis of performance.

The authors have successfully:

Enhanced the computational focus with a more structured analysis of AI techniques

Provided a clearer taxonomy of learning approaches (Figure 5)

Added detailed performance analysis and limitations (Table 4)

Clarified the unique contribution beyond existing reviews

Expanded critical discussion of methodological trends and future directions

Still I got several minor comments for the current paper:

1. I would like to first thank the author for providing extra introductions about how their work different from others. However, for my 3rd major concern, I would suggest the author add some context to discuss the specificity of leprosy care, possibly about why AI is important in leprosy care (compared to other disease) and what characteristics of leprosy made it suitable to be studied with AI.

2. The author has mentioned specific preprocessing challenges for leprosy image data or tabular data. If possible, the author could also discuss how inconsistent preprocessing affects comparison across studies.

3. The “RQ1: …”, “RQ2: …” headings still follow an empirical‑study format. PCB reviews typically use descriptive subheadings (e.g. “Leprosy thematic areas addressed”), without labeling them “RQ#.” Please adjust to match the journal’s review‐article style.

4. A few citations in the text (e.g. [37], [40], [41]) don’t appear in the reference list; please ensure every in‑text reference is included in the bibliography and vice versa.

5. Other tips that may help increase the overall quality of the manuscipt:

a)“MDT” appears in the Abstract without definition. Please spell out “multidrug therapy (MDT)” at first use.

b)Several captions (e.g. for Figures 3–7) are too concise. They should be fully self‑contained, describing what each axis represents, how categories are defined, sample sizes, and any color/shading conventions.

**Have the authors made all data and (if applicable) computational code underlying the findings in their manuscript fully available?**

Reviewer #1: Yes

Reviewer #2: None

Reviewer #3: Yes

Reviewer #4: Yes

PLOS authors have the option to publish the peer review history of their article (what does this mean?). If published, this will include your full peer review and any attached files.

Reviewer #1: No

Reviewer #2: No

Reviewer #3: No

Reviewer #4: No

**Figure resubmission:**
---

## [Decision Letter · Decision Letter 2]

3 Jun 2025

Dear Dr. Endo,

We are pleased to inform you that your manuscript 'On the usage of artificial intelligence in leprosy care: A systematic literature review' has been provisionally accepted for publication in PLOS Computational Biology.

Best regards,

Yang Lu, Ph.D.

Academic Editor

PLOS Computational Biology

Benjamin Althouse

Section Editor

PLOS Computational Biology

Reviewer's Responses to Questions

**Comments to the Authors:**

Reviewer #4: The author has addressed all my pervious comments.

**Have the authors made all data and (if applicable) computational code underlying the findings in their manuscript fully available?**

Reviewer #4: None

PLOS authors have the option to publish the peer review history of their article (what does this mean?). If published, this will include your full peer review and any attached files.

Reviewer #4: No

---

## [Editor Report · Acceptance letter]

PCOMPBIOL-D-24-01734R2

On the usage of artificial intelligence in leprosy care: A systematic literature review

Dear Dr Endo,

I am pleased to inform you that your manuscript has been formally accepted for publication in PLOS Computational Biology. Your manuscript is now with our production department and you will be notified of the publication date in due course.

With kind regards,

Anita Estes
